# A mitochondrial DNA hypomorph of cytochrome oxidase specifically impairs male fertility in *Drosophila melanogaster*

Maulik R Patel[1,2,3]*, Ganesh K Miriyala[1†], Aimee J Littleton[1†], Heiko Yang[4,5], Kien Trinh[6‡], Janet M Young[1], Scott R Kennedy[7], Yukiko M Yamashita[4,5], Leo J Pallanck[6], Harmit S Malik[1,2]*

[1]Division of Basic Sciences, Fred Hutchinson Cancer Research Center, Seattle, United States; [2]Howard Hughes Medical Institute, Seattle, United States; [3]Department of Biological Sciences, Vanderbilt University, Nashville, United States; [4]Life Sciences Institute, University of Michigan, Ann Arbor, United States; [5]Howard Hughes Medical Institute, University of Michigan, Ann Arbor, United States; [6]Genome Sciences, University of Washington, Seattle, United States; [7]Pathology, University of Washington Medical Center, Seattle, United States

**Abstract** Due to their strict maternal inheritance in most animals and plants, mitochondrial genomes are predicted to accumulate mutations that are beneficial or neutral in females but harmful in males. Although a few male-harming mtDNA mutations have been identified, consistent with this 'Mother's Curse', their effect on females has been largely unexplored. Here, we identify COII$^{G177S}$, a mtDNA hypomorph of cytochrome oxidase II, which specifically impairs male fertility due to defects in sperm development and function without impairing other male or female functions. COII$^{G177S}$ represents one of the clearest examples of a 'male-harming' mtDNA mutation in animals and suggest that the hypomorphic mtDNA mutations like COII$^{G177S}$ might specifically impair male gametogenesis. Intriguingly, some *D. melanogaster* nuclear genetic backgrounds can fully rescue COII$^{G177S}$-associated sterility, consistent with previously proposed models that nuclear genomes can regulate the phenotypic manifestation of mtDNA mutations.

*For correspondence: maulik.r.patel@vanderbilt.edu (MRP); hsmalik@fhcrc.org (HSM)

†These authors contributed equally to this work

Present address: ‡Biogen Idec, Cambridge, United States

## Introduction

The acquisition of the mitochondria by the ancestral eukaryote is one of the most remarkable instances of symbiosis in biology (*Sagan, 1967*; *Schwartz and Dayhoff, 1978*). This symbiosis gave the eukaryotic cell the ability to perform oxidative phosphorylation (*Williams et al., 2013*). In modern eukaryotes, oxidative phosphorylation is carried out by the electron transport chain, which comprises subunits encoded by both nuclear and mitochondrial genomes (mtDNA). Oxidative phosphorylation plays a fundamental role in many eukaryotes and is responsible for meeting most cellular energy demands. In humans, dysfunction of oxidative phosphorylation is associated with many diseases including cancer, diabetes, infertility, and neurodegenerative disorders (*Wallace, 2001*).

Despite the appearance of a symbiotic relationship, the evolutionary interests of mitochondria can be in conflict with those of nuclear genomes (*Partridge and Hurst, 1998*). This conflict arises from the differences in transmission between the two genomes. Whereas the nuclear genome is transmitted in a Mendelian fashion through both sexes, mtDNA is exclusively maternally inherited in most metazoans. Indeed, there are elaborate mechanisms to prevent the inheritance of sperm mitochondria (*Birky, 1995*; *Al Rawi et al., 2011*; *Sato and Sato, 2011*; *Zhou et al., 2011*; *DeLuca and*

**eLife digest** Cell compartments called mitochondria are responsible for producing much of the energy that animal and plant cells need. Most of the proteins in mitochondria are produced from genes found in another compartment called the nucleus. However, some mitochondrial proteins are made from genes found in the mitochondria themselves.

Unlike the genes in the nucleus, which animals and plants inherit from both their mother and father, the mitochondrial "genome" is only passed on along the female line. Therefore, males represent an evolutionary dead-end for mitochondrial genes. Evolutionary theory predicts that this should result in the evolution and spread of mutations that can be harmful to males, providing they do not reduce the ability of females to survive and reproduce. Although such 'male-harming' mutations have been well studied in plants, it is less clear how common they are in animals.

Patel, Miriyala, Littleton et al. used fruit flies as a model system to identify and characterize male-harming mutations in the mitochondrial genome. The experiments isolated a mitochondrial genome with a single mutation in a gene that encodes an enzyme called cytochrome oxidase II. The mutation is said to be "hypomorphic" because it lowers the activity of the gene. The fertility of male flies with this mutation rapidly declined as they aged. However, the mutation did not appear to lower the fertility of female flies. In fact, apart from the lower male fertility, the mitochondrial mutation did not seem to affect any other traits in males or females.

Further experiments revealed that this hypomorphic mutation specifically impairs the development of sperm. Patel, Miriyala, Littleton et al. also found that the effect of the mutation on the fertility of the males depended on the genes in the nucleus of their cells, as some nuclear genomes were able to partially or completely suppress the mutation. This supports previous findings that the effect of mitochondrial mutations in animals and plants may be complex and can be strongly influenced by the genes in their nucleus.

Patel, Miriyala, Littleton et al.'s findings suggest that sperm development is particularly susceptible to defects in mitochondria, and that hypomorphic mutations may represent a broader category of 'male-harming' mutations in animals. A future challenge will be to find out whether such mutations occur in humans and whether they are associated with infertility in men.

*O'Farrell, 2012*). Due to this uniparental inheritance, natural selection for mtDNA fitness can operate in females but not in males. As a result of this relaxed selection, 'male-harming' mtDNA mutations are expected to accumulate as long as they are beneficial, neutral or nearly neutral in females, a scenario referred to as the 'Mother's Curse' (*Frank and Hurst, 1996*; *Gemmell et al., 2004*). However, since decreased male fitness is detrimental for the evolutionary success of the nuclear genome, outbred populations subject to accumulation of male-harming mtDNA mutations can rapidly evolve nuclear genome-encoded suppressors to restore male fitness (*Rand et al., 2004*; *Dowling et al., 2008*). Thus, evolutionary theory predicts a conflict-driven molecular arms race between mtDNA and the nuclear genome.

These evolutionary predictions have spurred interest into the molecular basis of how mtDNA mutations can manifest as specifically 'male-harming' particularly since mitochondrial function would be expected to be similar in most organs of both sexes. One of the best-understood examples of such 'male-harming' mtDNA mutations are *cytoplasmic male sterility (cms)* mutants described in many plant species (*Budar et al., 2003*; *Chase, 2007*). Such *cms* mtDNA can induce male sterility in hermaphroditic plants that can be cross-pollinated. By reallocating resources for pollen production instead for production of seeds, *cms* mtDNA increase female fitness – and consequently increase their own fitness (*Budar et al., 2003*). Molecular characterization of *cms* mutants shows that their 'male-harming' (and female-beneficial) properties are caused by novel chimeric genes that are created via recombination between mtDNA molecules. Nuclear-encoded suppressors of *cms* belong to the pentatricopeptide repeat (PPR) protein family and use RNA editing to effectively neutralize these male-harming chimeric genes in mtDNA (*Budar et al., 2003*; *Chase, 2007*; *Castandet and Araya, 2012*).

Animal mtDNA genomes are relatively small compared to those in plants, and the presence of novel chimeric genes is extremely rare in animal mtDNA. Despite these limitations, there is evidence that mutations in animal mtDNA genes might be male-harming. For example, mtDNA haplogroup T in humans is associated with reduced sperm motility (*Ruiz-Pesini et al., 2000*). Polymorphisms in mtDNA that cause this phenotype are hypothesized to be significant contributors to untreatable male subfertility, known to affect 7–10% of men (*Baker, 1994*). Recent findings implicate mtDNA variation as a significant contributor to shorter male lifespans in many taxa (*Clancy, 2008*; *Camus et al., 2012*). Although the phenotypic consequences of these mtDNA mutations are severe in males, it remains unclear whether they are benign in females, *i.e.,* whether they are specifically 'male-harming' mtDNA mutations as predicted by the Mother's Curse hypothesis.

Indeed, in spite of the theoretical predictions, male-harming mtDNA mutations have proven difficult to detect in animals. Several factors account for this difficulty. First, in natural populations, there can be significant indirect selection on mtDNA to maintain male fitness when the fitness of females is dependent on related males (those with the same mtDNA) (*Wade and Brandvain, 2009*; *Hedrick, 2012*). Such indirect selection against male-harming mtDNA mutations may be especially stringent in populations with high rates of inbreeding (*Wade and Brandvain, 2009*; *Hedrick, 2012*). Such indirect selection for male fitness might keep male-harming mtDNA mutations at low frequencies in natural populations, making them difficult to detect without deep surveys of natural polymorphism.

A second factor that may impede the discovery of male-harming mtDNA mutations in natural populations is their suppression by nuclear genome-encoded loci, which are under stringent selective pressure to restore male fitness (*Rand et al., 2004*; *Dowling et al., 2008*). Effects of detrimental mtDNA variants could thus only be evident if they were separated from their nuclear-encoded suppressors. With this goal, many researchers have created combinations of nuclear and mtDNA genomes in *D. melanogaster*, either from different strains or species (*Rand et al., 2001*; *Maklakov et al., 2006*; *Rand et al., 2006*, *Dowling et al., 2007*, *2008*; *Montooth et al., 2010*; *Clancy et al., 2011*; *Correa et al., 2012*). Such combinations have revealed instances of dramatic incompatibility, even within species, which results in male but not female lifespan defects (*Camus et al., 2012*). Variation in mtDNA also affects sperm competitiveness in *D. melanogaster* (*Yee et al., 2013*). While these elegant experiments have revealed mtDNA-dependent affects on male lifespan and fertility, they did not comprehensively assess affects on female life-history traits; it is thus unclear whether these mutations are specifically male-harming. One further limitation of such mtDNA 'swaps' is that introduced mtDNA often have several co-inherited mutations. This linkage of different mutations makes it nearly impossible to implicate single mutations as being causal for the incompatibility or to understand their biological mechanism. An exception is a single point mutation in *D. melanogaster* mtDNA that has been causally linked to male sterility (*Clancy et al., 2011*). However, recent demonstration of this mtDNA haplotype's negative pleiotropic effects on fertility and aging, both within and between sexes suggests it is not a specifically male-harming mutation (*Camus et al., 2015*). Thus, we currently lack clear examples of specifically male-harming mtDNA mutations in animals. This has also left unanswered the larger question of molecular mechanisms by which mtDNA mutations can manifest their 'male-harming' phenotypes.

Here, we isolated a mtDNA variant that harbors a single non-synonymous change in subunit II of cytochrome c oxidase (COII$^{G177S}$). We find that COII$^{G177S}$ males have an age– and temperature–dependent decrease in fertility. This decrease in fertility is correlated with a drop in COII enzymatic activity, which remarkably does not result in defects in any other male or female phenotypic traits we measured. Cellular characterization reveals decreased sperm production and function in the mutant males. By combining evolutionary principles with detailed functional characterization, our study thus provides one of the clearest examples of a male-harming mtDNA mutation in animals, and provides insights into the stringent requirements for optimal mtDNA function in sperm development. We further show that the fertility defect in COII$^{G177S}$ males can be completely suppressed by diverse nuclear backgrounds derived from various *D. melanogaster* strains, as predicted by the theory of cyto-nuclear genetic conflict.

## Results

### Recovery of a male-harming mtDNA mutation

To isolate male-harming mtDNA mutations, we devised an experimental evolution strategy that was partly inspired by previous strategies to decouple male versus female evolution in *D. melanogaster* (*Rice, 1996*; *Rice, 1998*). In our strategy, *D. melanogaster* females are prevented from mating with their male siblings. Instead, virgin females are mated to naïve males from an external stock every generation (*Figure 1A*). This strategy is predicted to eliminate indirect selection against male-harming mtDNA mutations because the fitness of the population is not dependent on the males carrying the mtDNA mutations; mtDNA are effectively absolved from supporting male fitness (*Wade and Brandvain, 2009*). Moreover, continuous replacement of a large fraction of the nuclear genome every generation via crosses to external males practically eliminates the likelihood of evolution of nuclear suppressors during the course of the experiment. In contrast, selection of mtDNA function

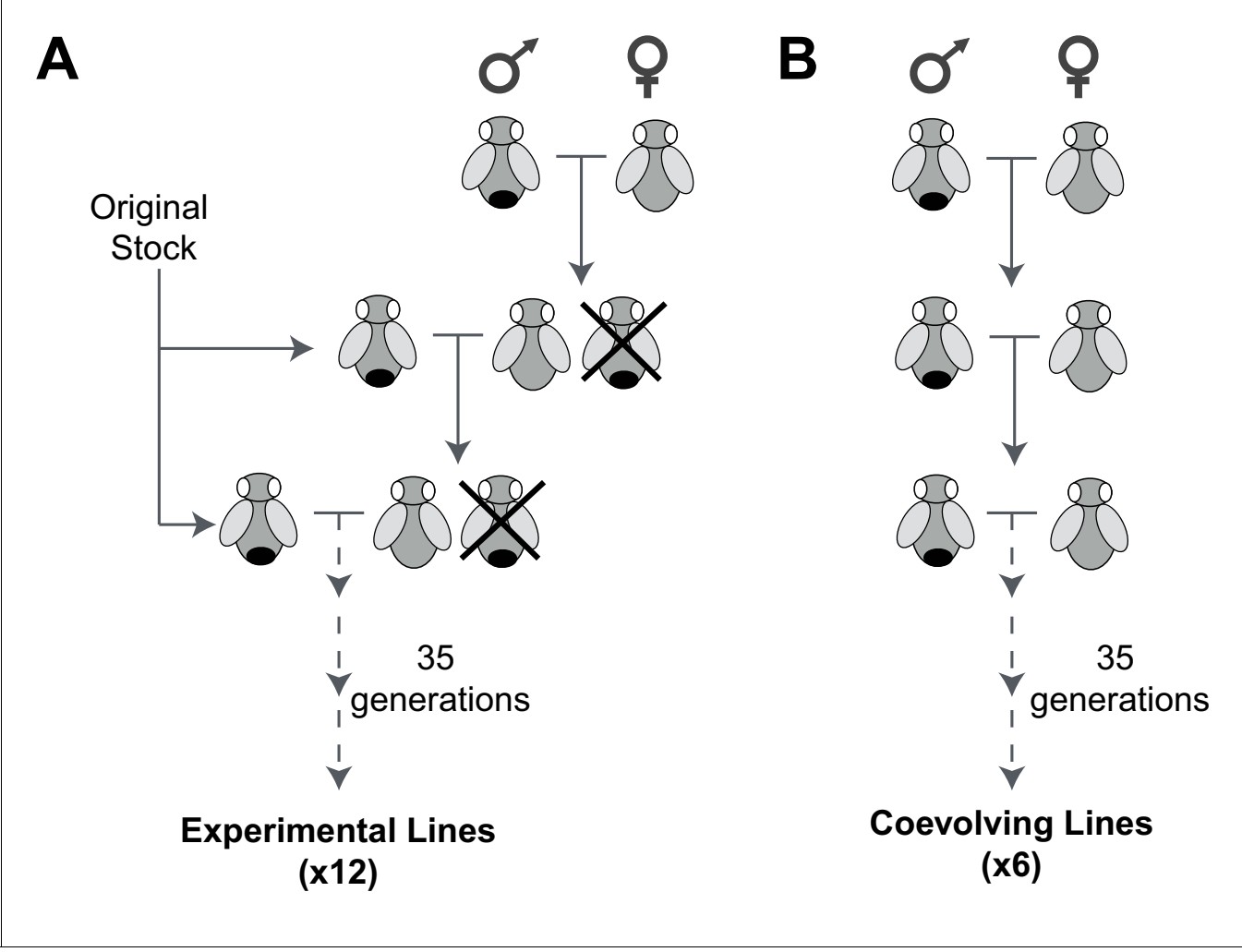

**Figure 1.** An experimental evolution strategy to recover male-harming mtDNA mutations. (**A**) In 12 *D. melanogaster* lines undergoing experimental evolution, 300 virgin female progeny in every generation were prevented from mating with sibling males (shown as being 'crossed out') and instead were mated with 100 males from the original stock. (**B**) In six lines undergoing coevolution, we allowed 300 virgin female and 100 male siblings to mate in every generation. Crosses for the 12 experimental and six coevolving lines were carried out for 35 generations.

The following figure supplement is available for figure 1:

**Figure supplement 1.** Status of Wolbachia infection in w1118 derived stocks.

for female fitness is stringently maintained over the entire course of the experiment. Thus, we anticipated this experimental evolution strategy might provide a permissive environment for accumulation and phenotypic manifestation of mtDNA mutations that are beneficial, neutral, or nearly neutral (only slightly deleterious) in females but cause defects in males.

From an 'original stock' of w1118 flies, we established 18 independent lines, each consisting of 300 females and 100 males. In 12 of these lines (experimental lines), virgin females were collected every generation and mated with males from the original stock (*Figure 1A*). Female flies were subjected to this mating scheme continuously for 35 generations over approximately 70 weeks. For the remaining 6 lines (coevolving lines), we allowed sibling males to mate with the females over the course of 35 generations (*Figure 1B*). Based on published mtDNA mutation rate of $6.2 \times 10^{-8}$ per site per fly generation (*Haag-Liautard et al., 2008*), we anticipated that multiple mutations per mtDNA would be sampled in 35 generations across the 18 lines. Any mtDNA mutations fixed at the end of the experiment must have either been pre-existing heteroplasmic mtDNA mutations that went to fixation (although the rate of fixation of such heteroplasmic mutations would be slow [*Solignac et al., 1987*]) or arose *de novo* in the experimental lines.

After 35 generations, we subjected flies to phenotypic analyses to assess whether any male-harming mutations had been sampled in the course of our experimental evolution strategy. To assess male viability, we first measured sex ratio in crosses between females with mtDNA subjected to experimental evolution (either 'experimental' or 'coevolving' lines) and control males (original *w1118* stock maintained as a separate population). We did not observe a change in male to female ratio in any of the 18 lines, suggesting that any mutations in mtDNA acquired over the 35 generations did not alter male viability (*Figure 2A*). Next, we measured male fertility by mating males derived from the starting, experimental, and coevolving lines with females from the original stock. For 17 of 18 lines, we did not detect any significant differences in male fertility. However, we found that males from experimental line 7 (hereafter referred to as 'EL7') sired significantly fewer progeny compared to males from all the other experimental, coevolved and original lines (*Figure 2B*). In contrast, EL7 females do not have any significant decline in fertility (*Figure 2C*). This suggested the possibility of a male-harming mtDNA mutation in the EL7 line that was not deleterious to female fertility. We investigated whether the decline in male fertility we observed manifested over the entire adulthood of EL7 male flies. We found no obvious reduction in male fertility over the first few days of adulthood (note that F1 progeny do not emerge until day 12) in EL7 compared to the other experimental or control lines. Instead, we observed that the reduction in male fertility was age-dependent (*Figure 2D*). Our findings are consistent with previous observations in which mitochondrial dysfunction is more severe in aged individuals (*Camus et al., 2012*; *Tower, 2015*).

## A single missense mtDNA mutation results in male fertility decline

To identify the putative mutation responsible for decreased male fertility, we performed whole-genome sequencing of the original and EL7 *D. melanogaster* lines using DNA isolated from a pool of 100 flies for each line. We were able to achieve average coverage of over 1000-fold for the entire mtDNA, except for the highly repetitive, AT-rich 'control' region (*Figure 3—figure supplement 1*). Analysis of these mtDNA sequences revealed a single missense mutation in EL7 mtDNA, resulting in a glycine to serine substitution at position 177 in subunit II of cytochrome c oxidase (COII^G177S), the fourth complex in the mitochondrial electron transport chain (*Figure 3A–B*). Interestingly, the COII^G177S mutation did not arise *de novo* during the course of our experiment. Although 98% of the sequencing reads from EL7 harbored the COII^G177S mutation, 59% of the reads from the original stock also corresponded to the mutant allele. Sanger sequencing confirmed this difference in proportion of wildtype and mutant mtDNA (*Figure 3—figure supplement 2*).

A mixture of wildtype and mutant mtDNA can exist in the same population in two ways. First, a population can consist of homoplasmic (carrying genetically identical mtDNA) wildtype and mutant individuals. Alternatively, the two mtDNA can be present in the same individual in a state of heteroplasmy. Sanger sequencing of single individuals from our original line revealed the presence of heteroplasmic individuals as well as flies that were homoplasmic wildtype and mutant (*Figure 3—figure supplement 3*). Therefore, 59% reflects the average combined frequency of the COII^G177S mutation from both heteroplasmic and homoplasmic mutant flies in the population.

The mutant allele's 59% frequency among reads from the original stock suggested that the COII^G177S mutation was present before initiation of the experimental lines. If this model is correct,

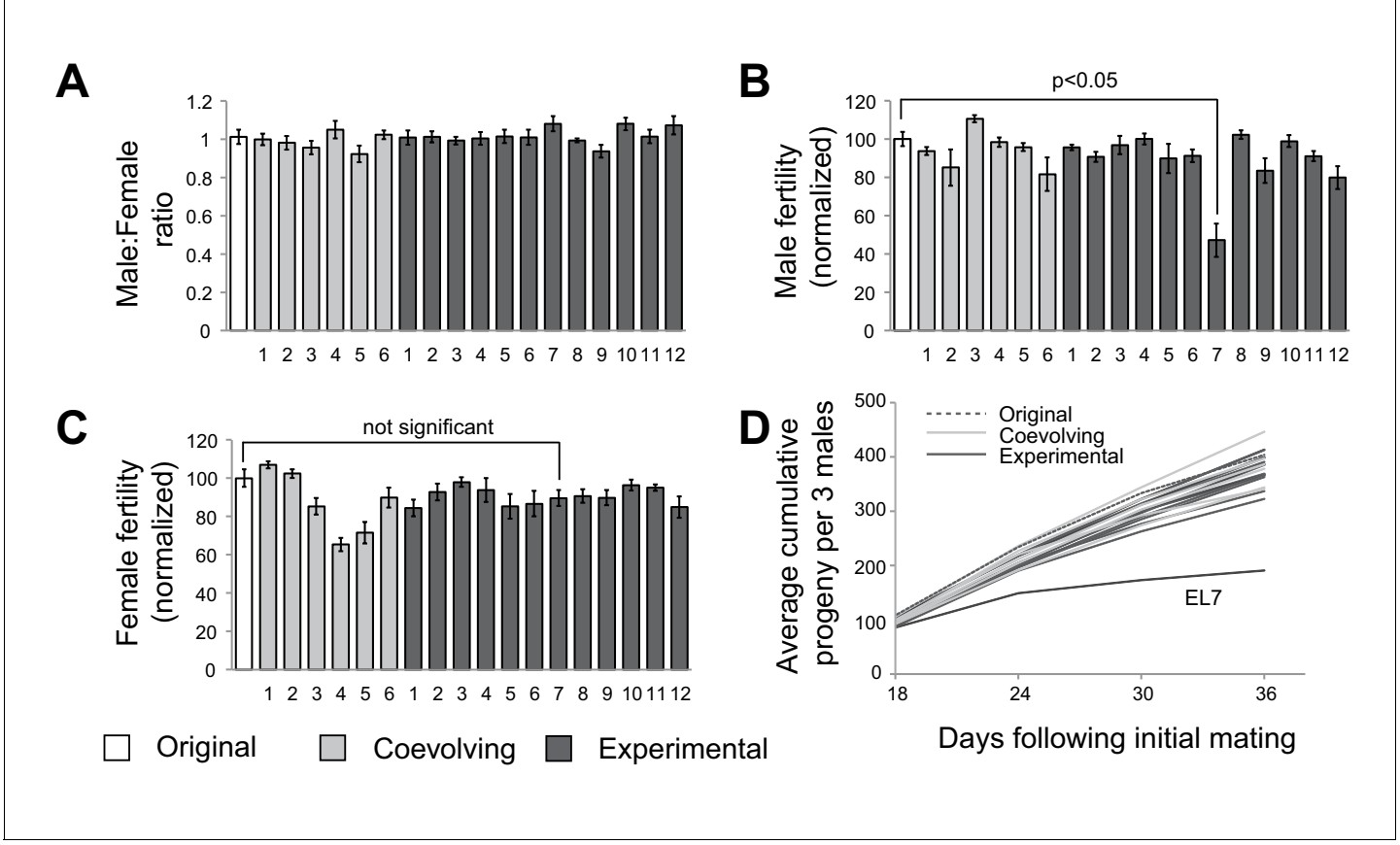

**Figure 2.** A single experimental line with male-specific fertility defects. (**A**) We measured average sex ratio of progeny by mating five females from each of the original, experimental, or coevolving lines, with three males from the original stock. Average sex ratio was found to be close to 50:50 in each of the lines, suggesting that there were no gross viability differences between male and female progeny in any of the lines. All error bars represent standard error of the mean. All experiments were done in replicates of 10 per group. (**B**) We measured male fertility by mating three males from each of the original stock, experimental, or coevolving lines with five females from the original stock, and calculating average number of resulting progeny. Male fertility is represented as a normalized percentage of progeny relative to the original stock. Only one line, experimental line 7 (EL7) showed significant reduction relative to the others. All error bars represent standard error of the mean. All experiments were done in replicates of 10 per group. (**C**) We measured female fertility by mating five females from each of the original, experimental, or coevolving lines with three males from the original stock, and calculating average number of resulting progeny. Like in (**B**), female fertility is represented as a normalized percentage of progeny relative to the original stock. EL7 female fertility is not significantly different from the original stock. All error bars represent standard error of the mean. All experiments were done in replicates of 10 per group. (**D**) To gain further insight into altered male fertility in EL7 (**B**), we plotted the cumulative number of progeny sired by (three) males as a function of time since initial mating (10 replicates per line). We find that the cumulative number of progeny sired by EL7 males is normal until day 18 but is subsequently significantly lower than for all other lines. This finding suggests an age-dependent decline in male fertility in the EL7 line.

we would expect that the COII$^{G177S}$ mutation should also be found in some of the other control and experimental lines in addition to EL7. Indeed, we found this to be the case (*Figure 3—figure supplement 2*). However, EL7 is unique in being the only population in which the COII$^{G177S}$ mutation is nearly fixed, whereas it varies from 0% to more than 50% in other experimental populations, which do not suffer any overt signs of male sterility at least at a population-level (*Figure 2B*, *Figure 3—figure supplement 2*). These data are consistent with the hypothesis that COII$^{G177S}$ mutation causes defects in male fertility but only when present at very high levels. Our findings are consistent with similar observations in pathogenic mtDNA variants in human disease (*Sobenin et al., 2014*).

Our finding that EL7 nearly fixed a mtDNA mutation that was already pre-existing provided another opportunity to test the association of the COII$^{G177S}$ mutation with decrease in male fertility. We reasoned that homoplasmic mutant males re-isolated from the original stock should also have reduced fertility despite the fact that these males have not undergone 35 generations of

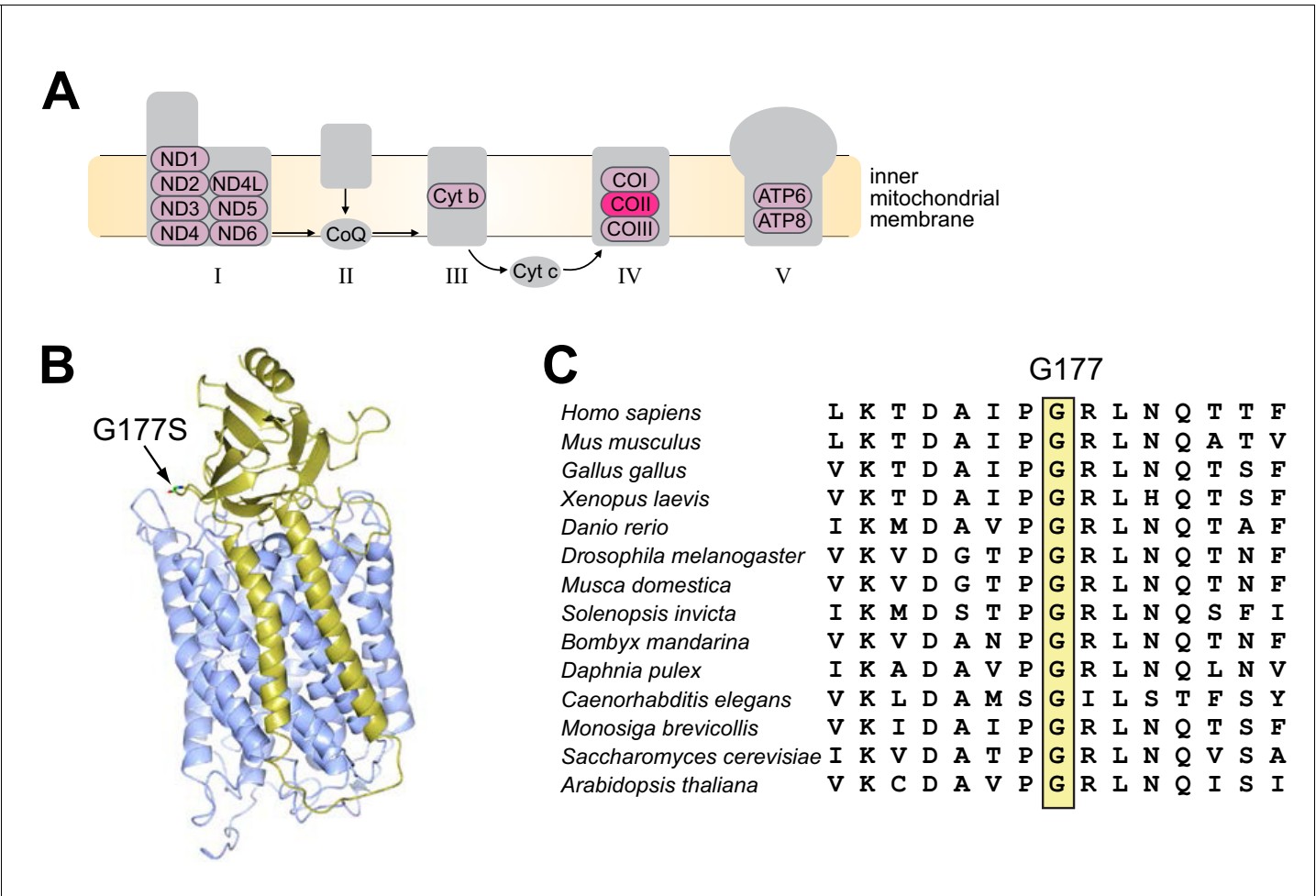

**Figure 3.** A single missense mutation (G177S) in subunit II of cytochrome c oxidase underlies lower male fertility in EL7. (**A**) The five complexes of the electron transport chain are schematized. These complexes are comprised of subunits encoded by both nuclear and mitochondrial genomes; the latter are indicated in pink. Subunit II of cytochrome c oxidase (COII) with the glycine to serine mutation at position 177 (G177S) in Experimental Line 7 is highlighted in bright pink. (**B**) Crystal structure of *Bos taurus* COII (in gold) in complex with COI (in blue) (PDB number: 2OCC). Glycine with its side chains at position 177 (G177) in COII is indicated. (**C**) Partial amino acid sequence alignment of COII from representative animal species highlights the conserved glycine residue at position 177 (boxed in yellow).

The following figure supplements are available for figure 3:

**Figure supplement 1.** Whole genome mtDNA sequencing coverage.

**Figure supplement 2.** COII[G177S] is present in many experimental and coevolving lines.

**Figure supplement 3.** COII[G177S] is present at variable levels in flies of the original w1118 stock.

experimental evolution like EL7. Consistent with this expectation, we were able to re-isolate homoplasmic COII[G177S] males, which sire fewer progeny compared to homoplasmic wildtype males derived from the same original stock (*Figure 4A*) (mutant and wildtype lines re-established from single females). These data demonstrate that the homoplasmic COII[G177S] mutation is necessary and sufficient to explain the observed decreased male fertility in EL7 males. These re-isolated COII[G177S] homoplasmic mutant mtDNA males also recapitulated the age-dependent decline in male fertility (*Figures 2D,4B*).

We characterized the nature of the heteroplasmy in the re-isolated COII[G177S] mutant mtDNA lines by sequencing a pool of COII[G177S] mutant mtDNA and a separate pool of re-isolated wildtype

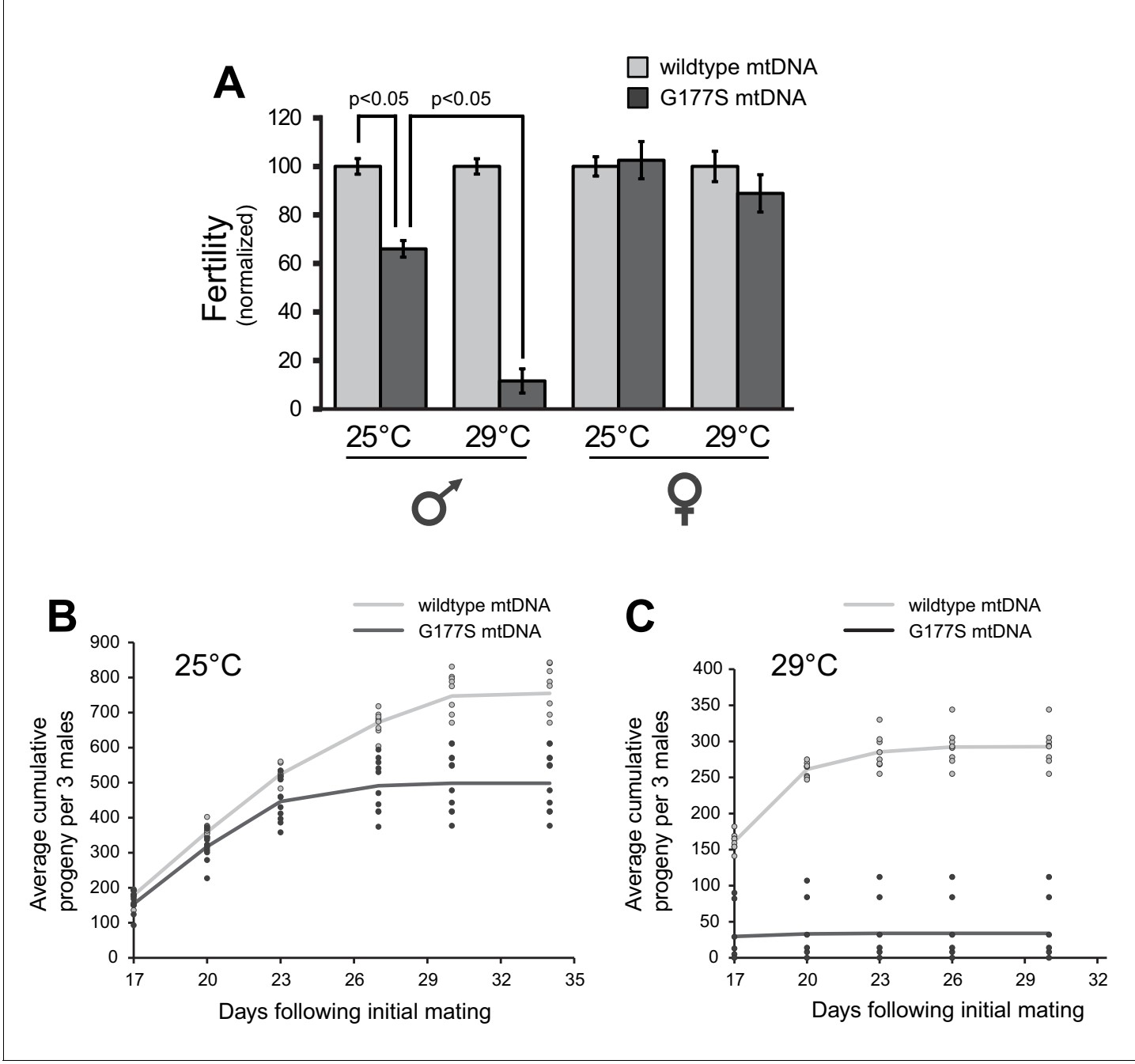

**Figure 4.** Males with COIIG177S mutation have decreased fertility at elevated temperatures. (A) To measure male fertility, we compared the fertility of males homoplasmic for either wildtype or COII[G177S] mutant mtDNA reestablished from the original stock. For each experiment, three males from homoplasmic stocks were mated with five females homoplasmic for wildtype mtDNA. Male fertility is presented as a normalized percentage of progeny produced, relative to the number produced by wildtype mtDNA males. COII[G177S] mutant mtDNA males produce fewer progeny at 25°C, but this is further reduced at 29°C. To measure female fertility, five females from homoplasmic wildtype or COII[G177S] mutant mtDNA stocks were mated with three males with wildtype mtDNA. Female fertility is presented as a normalized percentage of progeny produced, relative to wildtype mtDNA females. All error bars represent standard error of the mean. All experiments were done in replicates of 10 per group. (B) Number of progeny sired by wildtype or COII[G177S] mutant males at 25°C is cumulatively plotted as a function of time since initial mating (10 replicates per line). Actual number of cumulative progeny from each replicate is indicated by dots at given time points. (C) Number of progeny sired by wildtype or COII[G177S] mutant males at 29°C is cumulatively plotted as a function of time since initial mating (10 replicates per line). The reduction in male fertility is more significant at the higher temperature.

The following source data is available for figure 4:

*Figure 4 continued on next page*

*Figure 4 continued*

**Source data 1.** Near homoplasmy of wildtype and COII[G177S] mtDNA in re-isolated lines.

mtDNA adult fly heads using a Duplex Sequencing strategy (*Kennedy et al., 2014*) followed by hybrid capture to significantly enrich for mtDNA (*Figure 4—source data 1*). Under this strategy, mtDNA is sequenced at very high depth of coverage, labeling individual DNA molecules and sequencing each one multiple times in order to distinguish true mutations from sequencing errors. This strategy allows for sensitive evaluation of heteroplasmic mtDNA mutations. We found no evidence for wildtype mtDNA sequence in a pool of flies that we had re-isolated to enrich for the COII[G177S] mutant mtDNA (0 duplex consensus reads out of >7000). In contrast, we found no evidence for the COII[G177S] mutant mtDNA in flies (0 reads out of >7000) that we had re-isolated from the same original w1118 stock to enrich for wildtype mtDNA. We were cognizant that although the pooled COII[G177S] mutant flies showed no evidence of wild-type sequence, some heteroplasmy may nevertheless persist in individual flies. We therefore also sequenced 7 individual flies from the COII[G177S] mutant mtDNA pool to high depth of coverage. For six flies, we found no reads (out of >8,000) corresponding to wildtype mtDNA, whereas in one fly we uncovered seven out of >11,000 reads corresponding to wildtype mtDNA (<0.1%). We therefore conclude that the re-isolated strains are almost completely homoplasmic for either COII[G177S] mutant mtDNA or wildtype mtDNA. We therefore used these re-isolated strains for all subsequent phenotypic analyses.

Although we had set out to recover *de novo* mtDNA mutations that are male-harming using our experimental evolution strategy, we instead recovered what appears to be a pre-existing heteroplasmic mtDNA mutant, which fixed in only one of the 12 experimental lines. Therefore, we cannot attribute the isolation of this mutant to our scheme. Future work will be needed to determine the effectiveness of our experiment evolution scheme in recovering male-harming mutations, perhaps aided by experimentally increasing mtDNA mutation rates. For the rest of the manuscript, we focus on detailed characterization of the COII[G177S] mutation to determine its cellular consequences and to determine whether it is specifically male-harming.

## Decreased male fertility in COII[G177S] mutants is temperature dependent

All our analyses so far suggest that COII[G177S] is a specifically male-harming mtDNA mutation. Hence, we decided to further characterize the cellular and molecular basis underlying its detrimental effects. Previous experiments have shown that the phenotypic effects of mtDNA mutation can be exacerbated with increased stress, including higher temperatures (*Hoekstra et al., 2013*; *Chen et al., 2015*). We therefore investigated the effect of higher temperature on the fertility defect in COII[G177S] males. We found that COII[G177S] males are almost completely sterile when raised at 29°C instead of at 25°C (*Figure 4A*). Furthermore, we found that this defect manifests in both old and young males (*Figure 4C*). Thus, higher temperature provides a more sensitized condition to evaluate the various consequences of the COII[G177S] mtDNA mutation as previously observed (*Hoekstra et al., 2013*).

## COII[G177S] mutants have impaired cytochrome c oxidase activity

Next, we investigated the molecular consequences of the COII[G177S] mutation. COII is a subunit of cytochrome C oxidase (COX), which oxidizes the reduced form of cytochrome c. The glycine residue at position 177 is found in a loop of COII's structure where it comes in very close proximity to sub-unit I of the enzymatic complex (*Figure 3B*). Based on the fact that G177 is highly conserved across metazoans (*Figure 3C*), we hypothesized that the G177S mutation affects either an intrinsic function of COII or its interaction with other proteins in complex IV of the electron transport chain. To understand the biochemical consequences of the COII[G177S] mutation, we measured COX activity from whole fly lysates of flies grown at 25°C. Although the COX activity in COII[G177S] mutant flies was slightly lower than flies with wild-type mtDNA, this defect was not statistically significant, even in old flies (*Figure 5A*). We therefore measured COX activity from flies raised at 29°C where male fertility is most significantly impaired. Our analyses revealed an approximately 20% decrease in COX enzymatic activity in COII[G177S] mutants grown at 29°C in both male and female flies (*Figure 5B*). Given

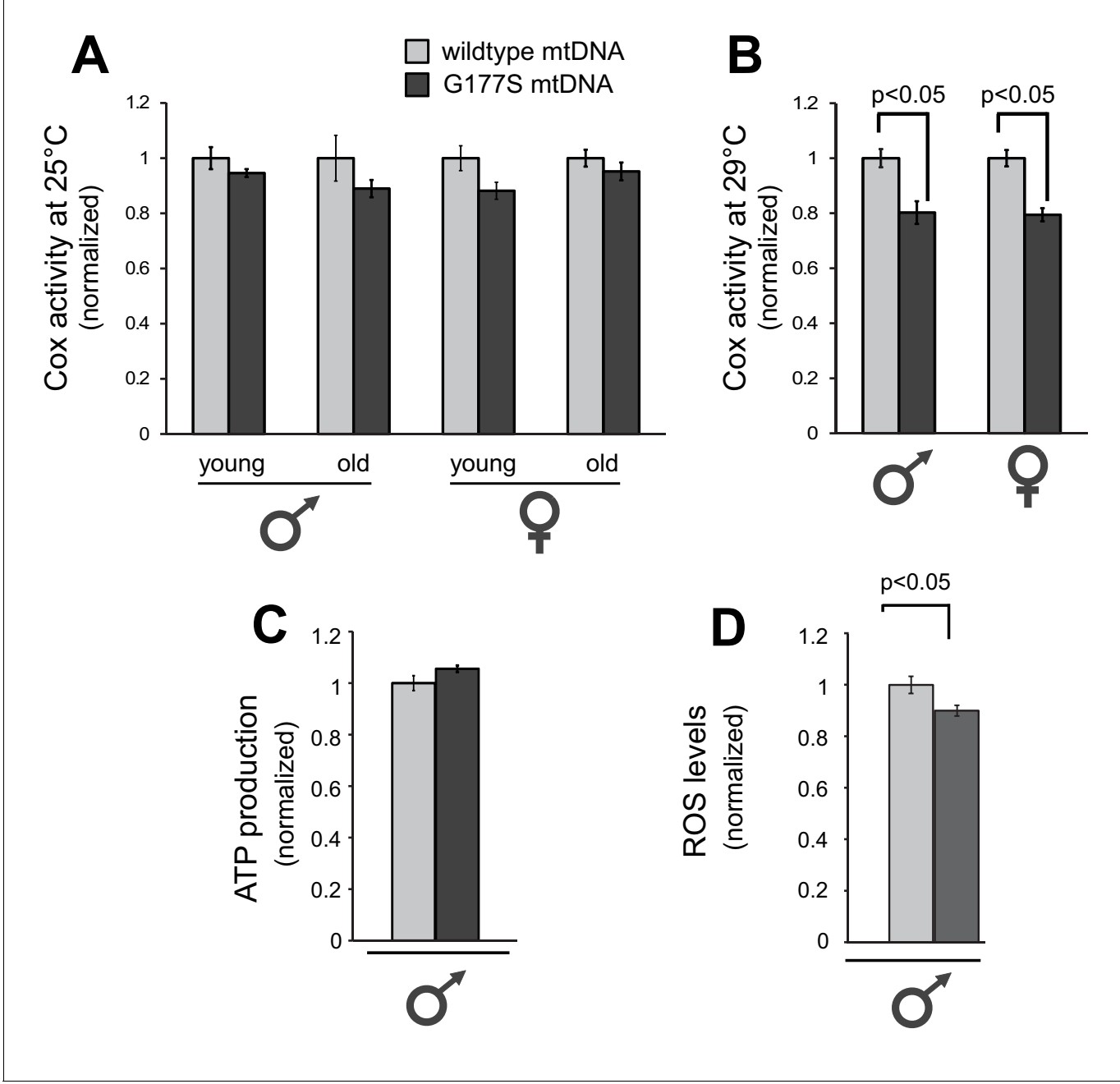

**Figure 5.** Biochemical consequences of COIIG177S mtDNA mutation. We measured COX activity from young (3–4 days) or old (21 days) flies raised at 25°C (**A**), or young flies raised at 29°C (**B**). (**C**) We measured ATP levels from 3–4 day old males raised at 29°C. (**D**) We measured ROS levels from 3–4 day old males raised at 29°C. 4–5 replicates per group for all experiments. Data is normalized to wildtype flies for each group in all experiments. Error bars represent standard error of the mean.

that the activity of the electron transport chain is coupled to ATP synthesis, we also measured ATP levels at 29°C. In contrast to COX activity, we did not observe a decrease in ATP levels in the COII[G177S] mutants (*Figure 5C*). However, we did observe a mild but significant decrease in reactive oxygen species (ROS) levels (*Figure 5D*). Thus, impaired COX activity and reduced ROS levels, but not ATP production, appear to correlate with loss of male fertility.

## The COII[G177S] mtDNA mutant specifically impairs male fertility

Our finding that COII[G177S] mutants result in lower COX activity of both males and females prompted us to evaluate several other phenotypes commonly seen in flies with mitochondrial impairment. Mitochondrial dysfunction is often associated with aging defects in many species (*Cho et al., 2011*; *Tower, 2015*). Naturally occurring variation in mtDNA is also known to affect aging in *D. melanogaster* (*Rand et al., 2006*; *Clancy, 2008*; *Camus et al., 2012*). We therefore assayed the lifespan of wildtype and COII[G177S] mutant males and females at both 25°C and 29°C. We did not observe any differences in lifespan, even at the sensitized higher temperature between wildtype and mutant flies in either males or females (*Figure 6*).

Sensitivity to physiological stress and neuronal function are also associated with mitochondrial dysfunction (*Fergestad et al., 2006*; *Pieczenik and Neustadt, 2007*; *Ugalde et al., 2007*; *Distelmaier et al., 2009*; *Celotto et al., 2011*). We therefore assayed heat intolerance, and 'bang sensitivity', a measure of neuronal dysfunction (*Burman et al., 2014*). There were no statistically significant differences in these traits in either males or females, even in aged flies grown at 29°C, the highly sensitized and susceptible condition (*Figure 6—figure supplements 1, 2*). Thus, despite the 20% reduction in COX activity in both males and females, the phenotypic consequences of COII[G177S] in *D. melanogaster* appear to be largely benign, with deleterious effects confined to male fertility alone.

## COII[G177S] impairs sperm development and motility

We next investigated the biological basis of the decreased male fertility in COII[G177S] mtDNA-bearing males. Male sterility could either result from defects in the ability of males to mate or can be a result of defective sperm development/function or a combination of both. In order to distinguish between these two possibilities, we took advantage of the fact that transfer of male accessory gland peptides during mating induces egg-laying in females (*Wolfner, 1997*). We find that mating wildtype females with COII[G177S] males induces egg-laying just as robustly as in females mated with males carrying wildtype mtDNA (*Figure 7A*). However, almost all of the eggs laid by females mated with COII[G177S] mutant males were unfertilized and failed to hatch (*Figure 7A*). Taken together, these data suggest that the decreased fertility in COII[G177S] males occurs due to defects in sperm development or function, and not due to mating ability. We therefore sought to determine the nature of cellular defects in COII[G177S] male testes.

Sperm development occurs within a cyst in which 64 sperm nuclei share a common cytoplasm. These sperm undergo individualization during the needle stage near the terminal epithelium of the testis. Examination of testes stained with the DNA marker DAPI revealed clear late needle-stage defects in COII[G177S] males. These mutant sperm failed to individualize properly and instead formed tangled clumps (*Figure 7B–C*). Consistent with this developmental defect, there is a significant reduction in the number of mature sperm that are stored in the seminal vesicles of COII[G177S] males raised at 29°C, as indicated by the decrease in the size of the seminal vesicles (*Figure 7D–F*). In addition, sperm that could be isolated from COII[G177S] males have reduced motility (*Videos 1,2*).

Although we used the higher temperature to provide a more sensitized assay to measure male fertility, we were aware of the possibility that the decreased fertility and the reduced seminal vesicle size could be solely the result of the elevated temperature (29°C). We therefore carried out similar experiments in aged males at 25°C, which also had displayed a reduction in fertility. Just like young COII[G177S] males at 29°C, we found that aged COII[G177S] males at 25°C also had reduced seminal vesicle size compared to aged males with wildtype mtDNA (*Figure 7—figure supplement 1*). In contrast, the seminal vesicle sizes of young males with wildtype mtDNA were the same, whether they were raised at 25°C or 29°C. Together, our results suggest that the decreased male fertility and reduced seminal vesicle size is not due to a confounding effect of higher temperature, but rather the result of the COII[G177S] mtDNA mutation.

Examining the male infertility phenotype further, we found that mitotracker green, a mitochondrial membrane potential-independent dye, preferentially stains immotile sperm whereas mitotracker CMXRos, a mitochondrial membrane potential-dependent dye, preferentially stains motile sperm in wild type testes. Sperm from COII[G177S] mutant males showed similar mitotracker green staining but significantly reduced CMXRos staining overall compared with wildtype sperm (*Figure 8*), suggesting that the lack of motility likely results from dysfunctional mitochondrial activity. Taken

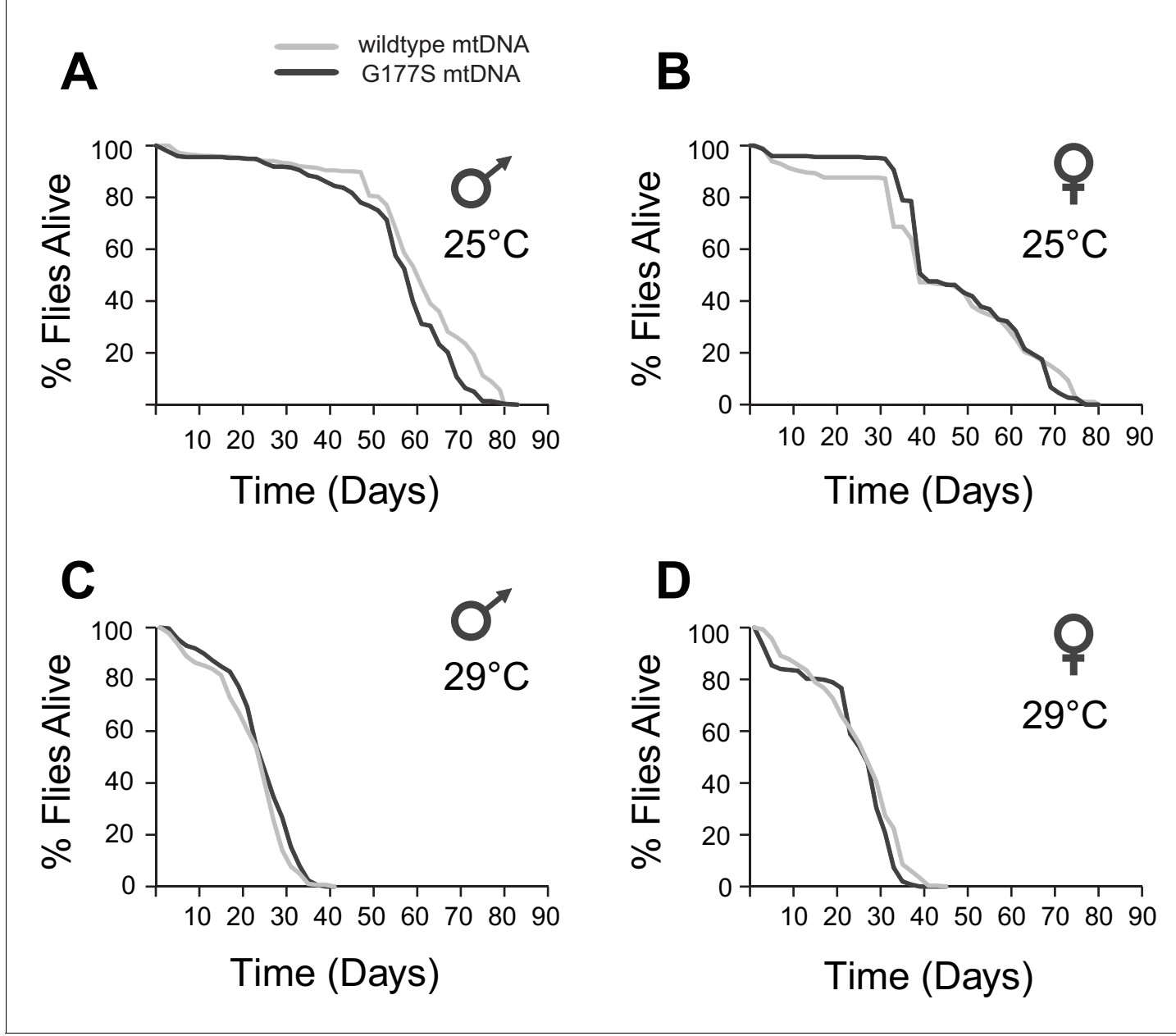

**Figure 6.** COIIG177S mtDNA mutation does not significantly affect male or female lifespan. We measured average lifespan of males and females at 25°C (**A,B**) or 29°C (**C,D**). N = 100 flies per group.

The following figure supplements are available for figure 6:

**Figure supplement 1.** COII[G177S] mutation does not affect heat tolerance of flies raised at 29°C.

**Figure supplement 2.** COII[G177S] mutation does not affect bang sensitivity.

together, these data demonstrate that decreased COX activity in COII[G177S] males correlates with the cellular defects in sperm development and motility in COII[G177S] mutants.

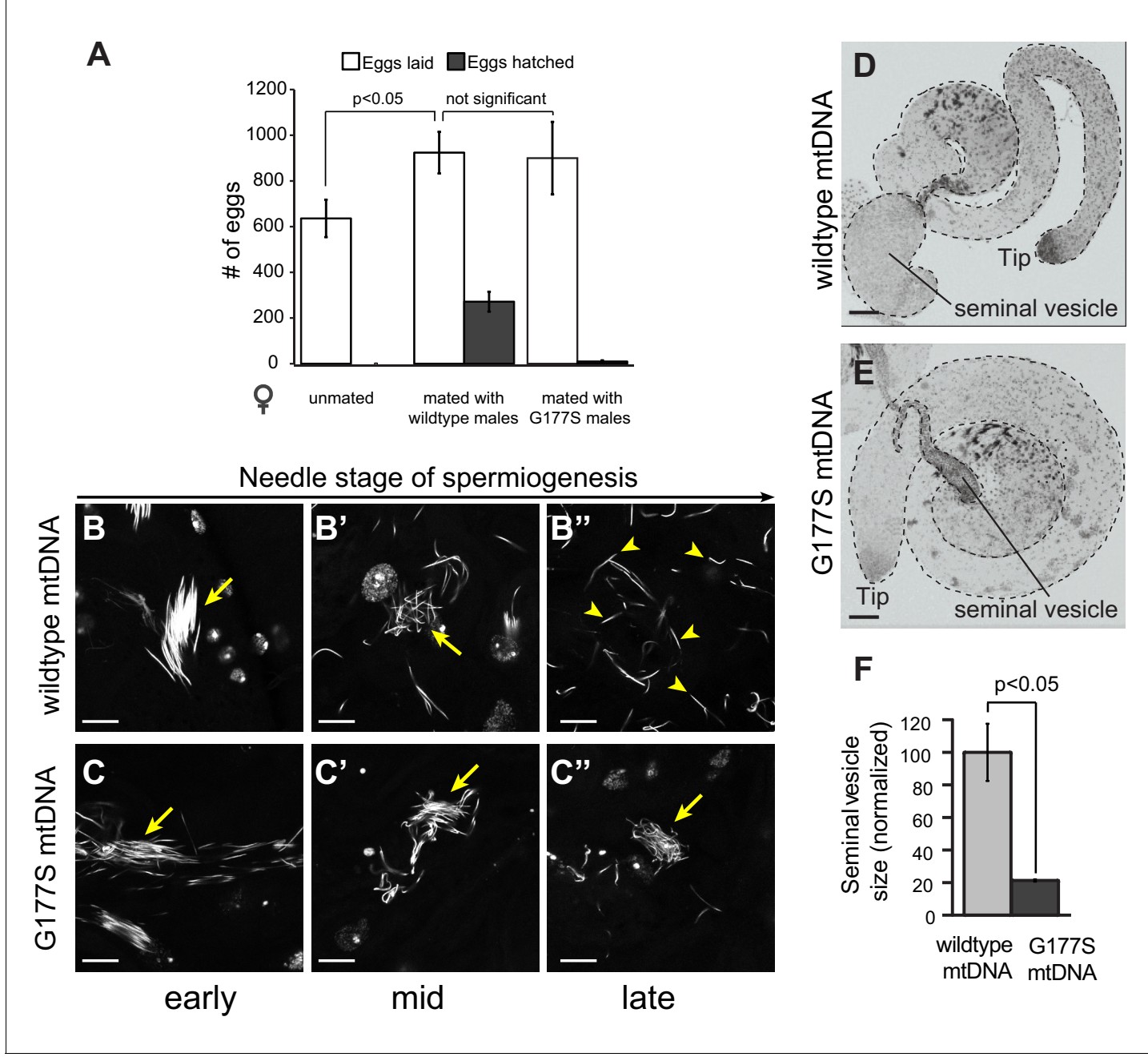

**Figure 7.** Normal mating but defective sperm development in COIIG177S mutant mtDNA males. (**A**) We measured whether the COII[G177S] mtDNA mutation affected the mating success of males. We measured eggs laid by virgin wildtype females that were either unmated, or mated with 2–5 day old wildtype, or COII[G177S] mutant males at 29°C. We determined the fraction of eggs hatched by counting unhatched eggs and larva 24 hrs after eggs were laid. All experiments were done in replicates of 6 per group. Error bars represent standard error of the mean. Our results show that the number of eggs laid after mating to wildtype mtDNA males is not significantly different from those mated to COII[G177S] mutant mtDNA males; in the latter case, most of the eggs are unfertilized and do not hatch. (**B,C**) We present maximum projection representative images of DAPI stained testis from 2–5 day old wildtype mtDNA male flies grown at 29°C at early (**B**), middle (**B'**), and late (**B''**) needle stage of sperm development. Note that the sperm are organized during early needle stage (arrow) and then break up into individual sperm by late needle stage (arrowheads). We also present maximum projection images of DAPI stained testis from COII[G177S] mutant mtDNA male flies grown at 29°C at early (**C**), middle (**C'**), and late (**C''**) needle stage of sperm development. Note also that the sperm in COII[G177S] mutant are 'clumped' and disorganized early in the needle stage and remain so through remainder of spermiogenesis (arrow). Scale bar, 20 μm. (**D,E**) Representative DAPI stained images of whole testis (outlined in dotted line) from 2–5 day old virgin wildtype mtDNA (**D**) and COII[G177S] mutant mtDNA males (**E**) raised at 29°C. For orientation, in both images, we identify the tip of the testis (where germ stem cells reside) as well as the seminal vesicle (the storage organ for mature sperm). Note the much smaller seminal vesicle size in the

*Figure 7 continued on next page*

*Figure 7 continued*

mutant males. Scale bar, 50 μm. (**F**) Quantification of the seminal vesicle size, as measured by cross-sectional area, normalized to wildtype. Average calculated from 5–7 testes. Error bars represent standard error of the mean.

The following figure supplement is available for figure 7:

**Figure supplement 1.** Aged males recapitulate the male sterility defects at 25°C.

## Nuclear genome variation can suppress the male-harming effects of COII^G177S mtDNA

Male-harming mtDNA mutations like COII^G177S are predicted to be detrimental for the evolutionary success of the nuclear genome. In the face of such mtDNA mutations, it is expected that nuclear genomes might have evolved suppressors to restore male fitness (*Rand et al., 2004*; *Dowling et al., 2008*; *Wolff et al., 2014*). To test this evolutionary hypothesis, we generated males heterozygous for varied nuclear genomes by crossing COII^G177S mtDNA-bearing females with males from a number of *D. melanogaster* strains, collected from different global populations. We then assayed the fertility of the resulting male progeny that carried the COII^G177S mtDNA but were heterozygous for the nuclear genome. To maximize the sensitivity of our assays, we performed these fertility assays in young males at 29°C. Surprisingly, we found that the nuclear genomes from many of the strains we tested were able to completely restore male fertility in COII^G177S males (*Figure 9A*, *Figure 9—figure supplement 1*).

Male fertility is highly sensitive to temperature, with different strains exhibiting different threshold of tolerance (*Rohmer et al., 2004*; *David et al., 2005*; *Hoekstra et al., 2013*). Hence, the restoration of fertility that we observe could simply reflect greater tolerance to high temperature independent of the COII^G177S mtDNA. To address this possibility, we measured total fertility of males until 21 days of age at 25°C. We found that the Oregon R nuclear background suppresses not only the young male sterility at 29°C but also male fertility at 25°C (*Figure 9B*, *Figure 9—figure supplement 1*). Thus, we conclude that the Oregon R nuclear background encodes a *bona fide* suppressor of COII^G177S mtDNA-mediated male infertility. We currently don't know the identity of the suppressor loci or whether these are specific only to the COII^G177S mtDNA mutation. For instance, these suppressors might employ a general mechanism that allows for rescue of male fitness defects caused by other mtDNA mutations. Under either scenario, we hypothesized that COII^G177S suppressor loci may be more abundant across *D. melanogaster* strains.

We used lines from the DGRP collection (*Mackay et al., 2012*), a set of fully sequenced inbred lines derived from a single natural population, to ask whether there is a lot of standing variation in the ability of nuclear genomes to suppress effects of the COII^G177S mutation. We found that DGRP line 861 was able to completely restore male fertility and most other lines were able to at least partially restore male fertility in males carrying COII^G177S mutation (*Figure 9c*, *Figure 9—figure supplement 1*). These data suggest that genetically dominant nuclear suppressors of the COII^G177S male fertility defects are widespread in natural populations. The different penetrance of the rescue also suggests that the suppression of COII^G177S associated male infertility is likely to involve multiple loci.

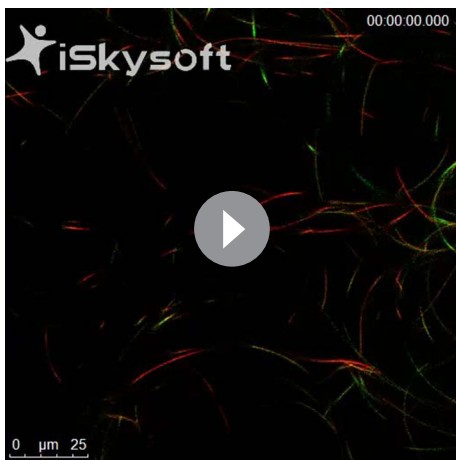

**Video 1.** Sperm motility assays in w1118 males carrying wildtype mtDNA. From a representative male grown at 29°C. Sperm stained with mitotracker Green (green), which stains immotile sperm and mitotracker CMS Rox (red), which stains motile sperm preferentially (related to *Figure 8*).

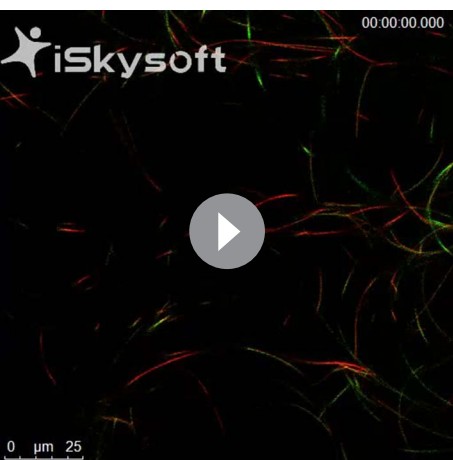

**Video 2.** Sperm motility assays in w1118 males carrying COIIG177S mutant mtDNA. From a representative male grown at 29°C. Sperm stained with mitotracker Green (green), which stains immotile sperm and mitotracker CMS Rox (red), which stains motile sperm preferentially (related to *Figure 8*).

We examined in more detail the cellular basis of 'rescued' male fertility in OregonR-w1118 heterozygous males bearing the COII^G177S mtDNA. We found that sperm development and motility were partially rescued in young males raised at 29°C (*Figure 10A–E*, *Figure 10—figure supplement 1*, *Video 3*) and aged males at 25°C (*Figure 7 —figure supplement 1*), consistent with the restoration of fertility that we previously observed (*Figure 9A,B*). We also evaluated the biochemical consequence of such nuclear suppression of COII^G177S mtDNA's male sterility to distinguish between two possible mechanisms of suppression. Under the first alternative, restoration of fertility in COII^G177S males might be due to higher baseline COX activity in the suppressor background, which can compensate for the hypomorphic COII^G177S and therefore be sufficient for sperm development and motility. Under this scenario, there would still be a difference in COX activity between wildtype and COII^G177S flies. A second alternative is that there is no difference in COX activity in flies carrying wildtype or COII^G177S mtDNA. When we measured COX activity in the suppressed nuclear background, we found the second scenario to be true *i.e.*, COX activity is the same in wildtype and COII^G177S mtDNA-bearing males (*Figure 10F*). This result was also observed for three additional suppressor lines we tested (*Figure 10—figure supplement 2*). These data suggest that in the suppressed background, COII^G177S mtDNA is as functionally effective as wildtype mtDNA and is indicative of a compensatory mechanism that does not rely simply on higher baseline levels of COX activity.

## Discussion

Evolution by natural selection rests on a simple but fundamental premise that the phenotypic manifestation of a genotype occurs in the same individual that transmits it to the next generation. Remarkably, the mitochondrial genome (mtDNA) does not abide by this rule since it is inherited and phenotypically manifests in both sexes, but is exclusively transmitted through females in most animals and plants. Thus, the phenotypic manifestation of mtDNA is decoupled from its transmission in males. Consequently, natural selection is ineffective at directly removing mtDNA mutations that are specifically harmful to males but not females. While the mtDNA phenotype/transmission decoupling clearly predicts existence of male-harming mutations (Mother's Curse) (*Frank and Hurst, 1996*; *Gemmell et al., 2004*), a systematic study of such mutations and their underlying mechanisms is hampered by the relative paucity of such mutations in animals.

We devised an experimental evolution scheme intended to absolve mtDNA of supporting male function while both maintaining selection on female mtDNA function as well as reducing the possibility of nuclear suppression of male-harming mtDNA. While we did recover a *bona fide* male-harming COII^G177S mtDNA mutation via this scheme, we were subsequently surprised to discover that this was a pre-existing mtDNA mutation that was already present at moderate frequencies in our starting laboratory strain of w1118. It is possible that the experimental evolution scheme might have created the permissive conditions that allowed this mtDNA mutation to rise to fixation in one strain. However, the fact that it did not rise to fixation in the other experimental populations means that we cannot conclude that the experimental strategy itself was causally important for the fixation of this male-harming mtDNA mutant. Future work on fine-tuning the strategy can address its effectiveness. For example, the scheme can be improved by increasing mtDNA mutation rates to improve sampling of *de novo* mutations, perhaps by employing a mutator mtDNA polymerase before initiating the experimental evolution scheme (*Trifunovic et al., 2004*). Alternatively, random chemical

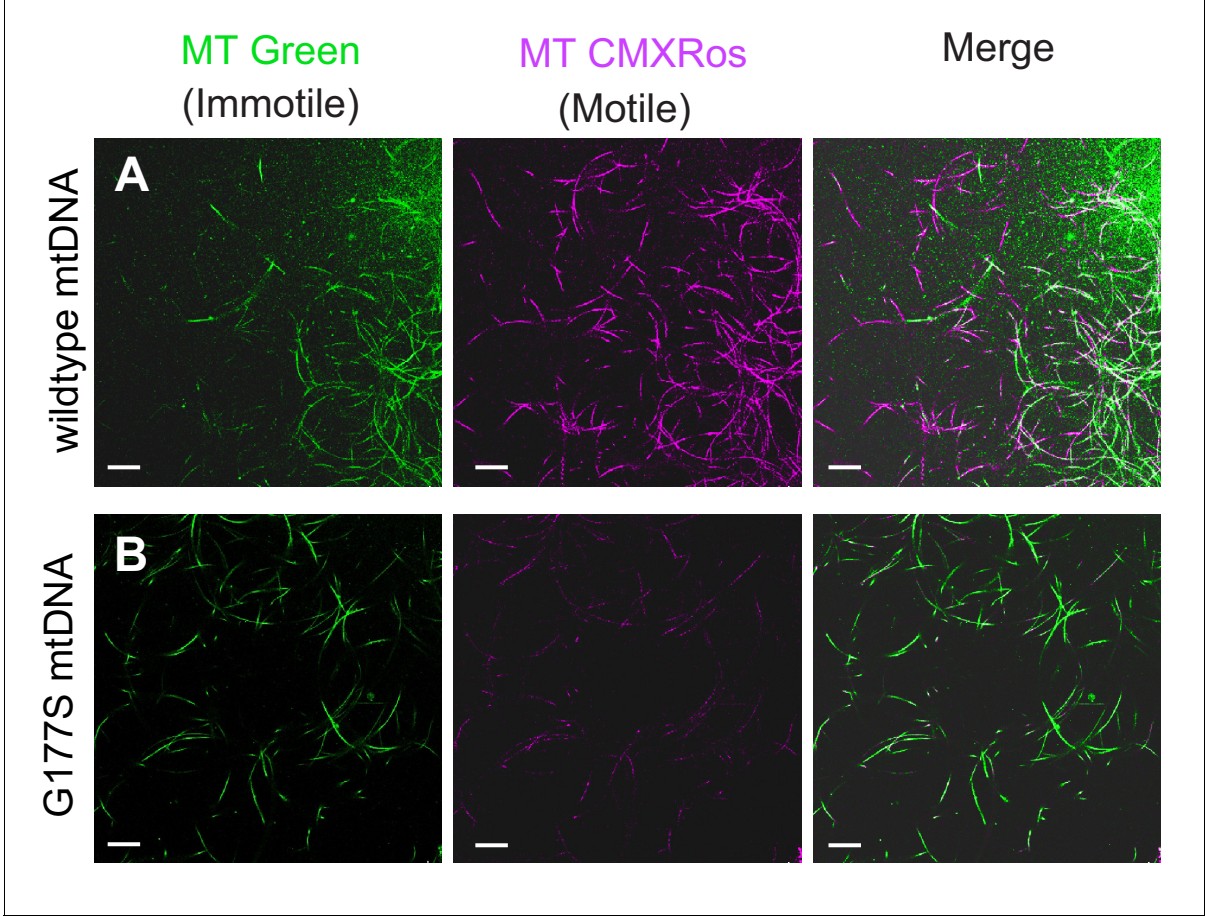

**Figure 8.** Lower motility of sperm from COII[G177S] mutant mtDNA males. Sperm from wildtype mtDNA (**A**) and COII[G177S] mutant mtDNA (**B**) males raised at 29°C stained with mitotracker Green (green), which stains immotile sperm and mitotracker CMS Rox (magenta), which stains motile sperm preferentially. Scale bar, 20 µm.

mutagenesis might also be readily applied here in our scheme. Although such chemical mutagenesis will affect both mtDNA and nuclear genomes initially, backcrossing to 'original stock' males in our scheme will rapidly replace most mutated nuclear genes with wildtype versions, while enabling the transmission of mtDNA mutations that do not affect female fitness. Finally, combining the crossing scheme we propose with the targeted restriction endonuclease strategy to generate mtDNA mutations (*Xu et al., 2008*) might be a facile means to discriminate specifically 'male-harming' mtDNA mutations from those that grossly impair mtDNA function.

Our phenotypic assays did not provide any evidence for the COII[G177S] mutation being beneficial in females. Furthermore, the mutation rose to high frequency in only one out of the twelve experimental lines, suggesting that the mutation is likely neutral or nearly neutral in females and therefore became nearly fixed through drift rather than selection. These data are consistent with the 'selective sieve' model, which predicts that even mutations that are neutral or nearly neutral in females but deleterious in males might be common because they cannot be effectively removed by natural selection (*Innocenti et al., 2011*). Our survey of the sequenced genomes from a large panel of 290 *D. melanogaster* strains and isofemale lines did not reveal any other lines besides our w1118 stock that harbored COII[G177S] (*Richardson et al., 2012*). Thus, COII[G177S] might be rare in natural populations.

The homoplasmic COII[G177S] mutation results in a 20% decline in COX activity at 29°C. Remarkably, we observed no phenotypic effects of this decline in COX activity on other phenotypic traits in both males and females, even at high temperatures. The COX activity decline specifically impaired sperm production and function. This establishes COII[G177S] as one of the first *bona fide* male-harming

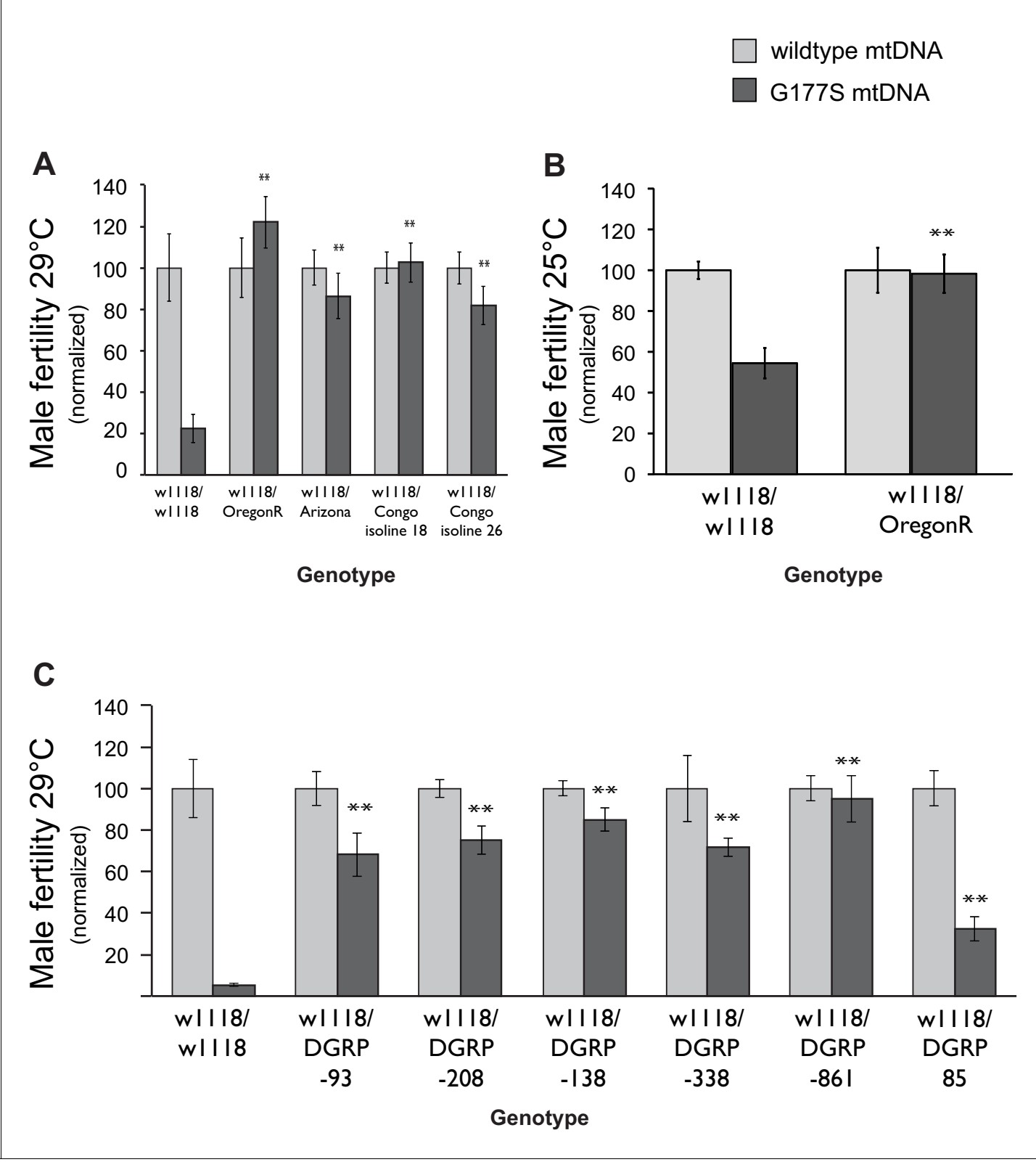

**Figure 9.** Dominant suppression of COII[G177S] associated male fertility defects by diverse nuclear genome backgrounds. (**A**) We measured male fertility of heterozygous males derived from crosses of w1118 females (carrying wildtype or COII[G177S] mutant mtDNA) to males from a variety of different nuclear backgrounds (*i.e.,* Oregon R, Arizona, Congo isoline 18, Congo isoline 26). In each case, three heterozygous males were mated with 5 w1118 females at 29°C (10 replicates per group). Male fertility was measured as average number of resulting progeny, normalized as a percentage of progeny

*Figure 9 continued on next page*

*Figure 9 continued*

of corresponding males with wildtype mtDNA. Raw progeny numbers are given in *Figure 9—figure supplement 1*. Most nuclear backgrounds show mild to complete suppression of the male fertility defects observed in w1118/w118 males carrying COII$^{G177S}$ mutant mtDNA (**p<0.05). Error bars represent standard error of the mean. (B) We examined lifetime male fertility in the w1118/Oregon R background at 25°C (note that male fertility is generally higher at lower temperatures) to quantify the suppression of male sterility. (C) We measured male fertility of heterozygous males derived from crosses of w1118 females (carrying wildtype or COII$^{G177S}$ mutant mtDNA) to males from six DGRP strains, at 29°C (10 replicates per group) as in (A). At least five of the tested DGRP strains show partial to complete suppression of male sterility.

The following figure supplement is available for figure 9:

**Figure supplement 1.** Dominant suppression of COII$^{G177S}$-associated male fertility by diverse nuclear genome backgrounds.

mtDNA mutations in animals. We considered the possibility that the phenotypic effects we have observed on male fertility are attributable to linked but unassayed changes in the 4-kb long AT-rich D loop control region of mtDNA. Indeed, the D-loop is one of the most rapidly evolving segments of mtDNA in *D. melanogaster*, but its highly repetitive nature challenges sequence characterization. However, our analyses allow us to directly implicate the COII$^{G177S}$ mtDNA mutation as being causally linked to the phenotypes we have observed. Not only do we observe a perfect correlation of the male fertility phenotype with the reduction of COX activity, but we also observe a near complete restoration of male fertility upon replenishment of COX activity in the suppressor strains.

Surprisingly, the decline in COX activity in COII$^{G177S}$ flies was not associated with a corresponding decline in ATP production. Since we were unable to reliably measure either COX activity or ATP production in testes, we cannot rule out the possibility that ATP production is specifically impaired in the male germline. However, our findings suggest a different molecular consequence of lowered COX activity may be responsible for defects in sperm development; we did observe decreased reactive oxygen species (ROS) production in COII$^{G177S}$ mutants. Our findings suggest that perhaps alterations in ROS levels might underlie the defects we see in sperm production and function. ROS have been shown previously to act as a signaling molecule to control the cell cycle checkpoint as well as the differentiation of hematopoietic progenitors during development in *D. melanogaster* (*Owusu-Ansah et al., 2008*; *Owusu-Ansah and Banerjee, 2009*). These data leave open the possibility of ROS similarly acting as a signaling factor during sperm development.

We hypothesize that male fertility may be a common target of male-harming mtDNA mutations because the reproductive tissues are highly sexually dimorphic. This hypothesis is consistent with previous data, which showed that naturally occurring variation in *D. melanogaster* mtDNA largely affects expression of male-expressed genes in the testis and the accessory gland (*Innocenti et al., 2011*). Two separate mutations in mtDNA are known to cause male sterility in *D. melanogaster* (*Xu et al., 2008*; *Clancy et al., 2011*). A single amino acid mtDNA mutation (A278T) in Cytochrome B of complex III renders males sterile; the primary defect appears to be at the level of spermatid individualization (*Clancy et al., 2011*). In contrast, a single amino acid mutation in Cytochome Oxidase I (R301L) causes male sterility primarily due to a sperm storage defect. However, the effects of these mutations on female fitness have not yet been comprehensively addressed so we cannot attribute them to be specifically 'male-harming'.

How might mtDNA function be different in testis compared to other tissues? It is possible that the testis has a differential requirement for COX activity. According to this hypothesis, although all tissues suffer from the relative reduction in COX activity, testis function is specifically impacted because it has a lower threshold of tolerance for a relative reduction in COX activity. In particular, COX activity might be required to facilitate the dramatic morphological changes that mitochondria undergo during sperm development in *D. melanogaster* (*Fuller, 1998*). If this hypothesis is correct, mildly hypomorphic mutations like COII$^{G177S}$ might generally impair male fertility and thus represent a common mechanism of male-harming mtDNA mutations. Alternatively, the existence of a number of nuclear-encoded testis-specific components of the electron transport chain, including subunits of cytochrome C oxidase, provide a number of interacting partners that might exhibit testis-specific genetic incompatibility with the COII$^{G177S}$ mutation (*Tripoli et al., 2005*; *Gallach et al., 2010*). Since we were unable to reliably measure COX activity in dissected testes of wildtype and mutant flies, we

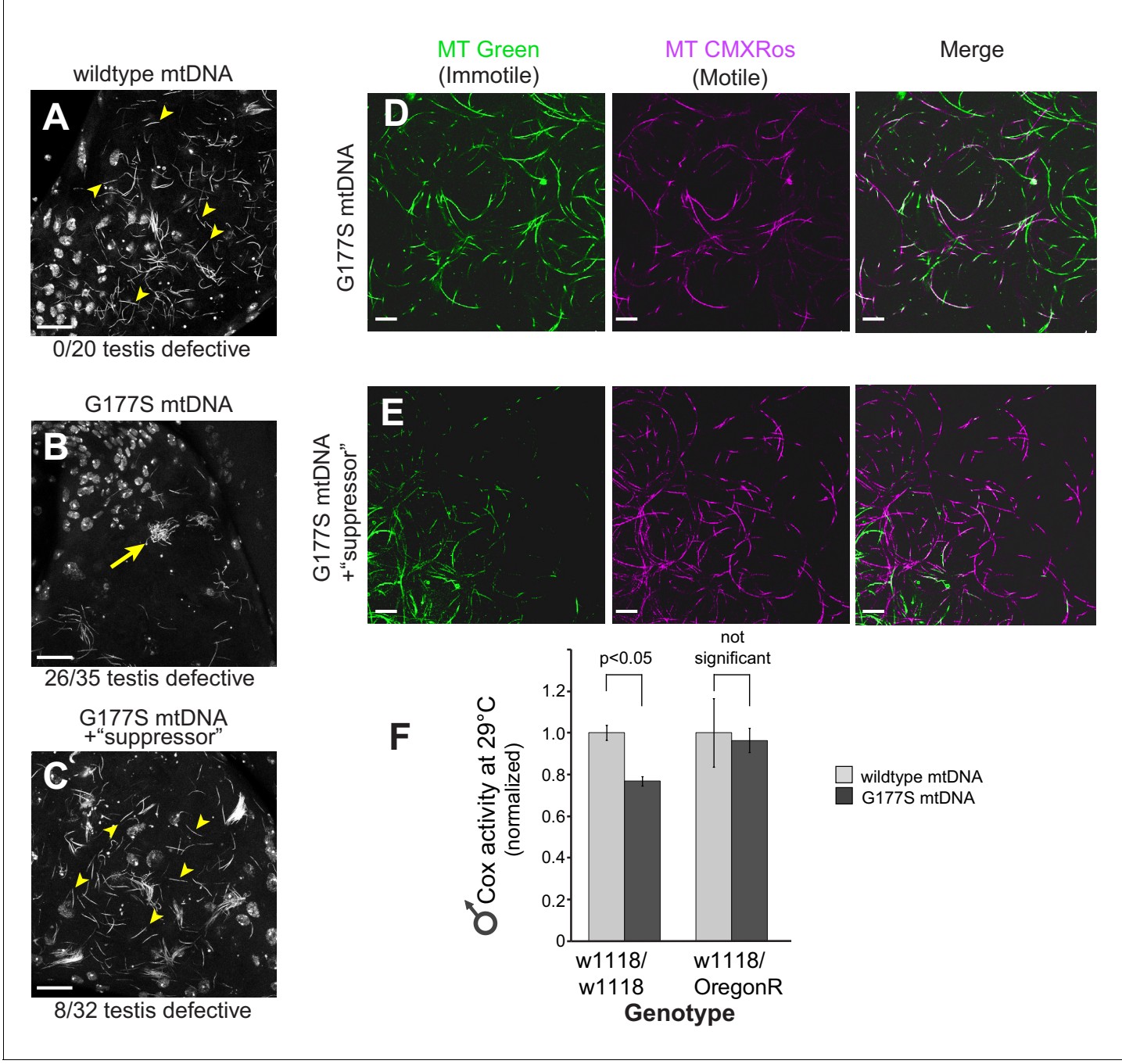

**Figure 10.** w1118/OreR heterozygous males carrying COII[G177S] mutant mtDNA show rescued sperm development and COX activity. Representative DAPI stained images of the late needle stage of sperm development from w1118/w1118 males carrying either wildtype mtDNA (**A**), or COII[G177S] mtDNA (**B**) compared to heterozygous w1118/OreR males carrying COII[G177S] mtDNA (**C**). Numbers of examined males with defects in sperm development are indicated. Arrow points to 'clumped' sperm. Note the presence of many individualized sperm (arrowheads) in A and C but few in B. Scale bar, 20 μm. We stained sperm from homozygous w1118/w1118 males (**D**) and w1118/OreR heterozygous males (**E**), both carrying COII[G177S] mutant mtDNA with mitotracker Green (green) and mitotracker CMS Rox (magenta), which stain immotile and motile sperm respectively. All flies were raised at 29°C. Scale bar, 20 μm. (**F**) COX activity measured from young (3–4 day old) w1118/w1118 males and w1118/OreR heterozygous males both carrying COII[G177S] mutant mtDNA. All flies were raised at 29°C. Data is normalized to flies with wildtype mtDNA in the corresponding nuclear background. Error bars represent standard error of the mean.

The following figure supplements are available for figure 10:

**Figure supplement 1.** Seminal vesicle size is restored in aged 'suppressor' backgrounds in males containing COII[G177S] mutant mtDNA raised at 25°C.

*Figure 10 continued on next page*

*Figure 10 continued*

**Figure supplement 2.** COX activity is specifically restored in 'suppressor' nuclear backgrounds.

are unable to address whether the decrease in COX activity is more severe in testes than other tissues.

Our study has identified one mtDNA mutation, COII[G177S], whose phenotypic effects appear to be restricted to the male germline. However, other forms of sexual dimorphism might lead to sex-specific phenotypic manifestations of other mtDNA mutations beyond reproductive tissues. For instance, there is already evidence that mtDNA might harbor mutations that affect aging and life-span, which is highly dimorphic between the sexes (*Camus et al., 2012*). At the molecular level, COX activity was shown to be differentially affected in male versus female flies based on their mtDNA haplotype (*Sackton et al., 2003*). Furthermore, almost 90% of the transcriptome in *D. melanogaster* is differentially expressed between the sexes (*Ayroles et al., 2009*). Amongst those differentially expressed genes, male-biased transcripts are enriched for mitochondrial energy metabolism, thus providing ample opportunities for sex-specific effects of mtDNA (*Ayroles et al., 2009*). Consistent with the nuclear background exerting an effect on the phenotypic manifestation of mtDNA mutants, we found that the COII[G177S] -mediated sterility was suppressed in many of the *D. melanogaster* nuclear backgrounds tested. Thus, our study not only shows the existence and biological basis of a male-harming mutation, but also demonstrates importance of the intimate relationship between nuclear and mtDNA genomes for the phenotypic manifestation of mtDNA mutations.

Our finding that the phenotypic consequences of a specific male-harming mtDNA mutation are dependent on the nuclear genome also re-emphasizes the exciting possibility of uncovering the genetic basis of nuclear-mitochondrial incompatibilities and the molecular basis of buffering by nuclear genomes. Such buffering provides an important insight into the genetic basis of human mtDNA disease in which male-harming mtDNA mutations might initially increase in frequency in a 'suppressor' background while avoiding any negative fitness consequences. The phenotypic consequences of male-harming mtDNA mutations would only be exposed in a naïve nuclear genetic background upon introduction into new populations or even (in the laboratory setting) new species. Such nuclear-mitochondrial incompatibilities have been proposed to potentially serve as the basis of speciation (*Burton and Barreto, 2012*). Although only a few such nuclear-mitochondrial incompatibilities have been mapped in molecular detail (*Ellison and Burton, 2006*; *Lee et al., 2008*; *Chou et al., 2010*; *Clancy et al., 2011*; *Meiklejohn et al., 2013*), studies in which mtDNA swaps were made within and between species have revealed that many such incompatibilities exist (*James and Ballard, 2003*; *Sackton et al., 2003*; *Ellison and Burton, 2006*; *Ballard et al., 2007*; *Lee et al., 2008*; *Montooth et al., 2010*). Future genetic mapping experiments will help identify the nature and mechanism of nuclear suppression of COII[G177S] and reveal whether they are specific or general suppressors of COII[G177S] and other male-harming mtDNA mutations. Strategies such as the one we have outlined here, together with new advances in manipulating mtDNA genomes (*Xu et al., 2008*), provide an exciting means to uncover the dark side of one of the most ancient symbioses on the planet.

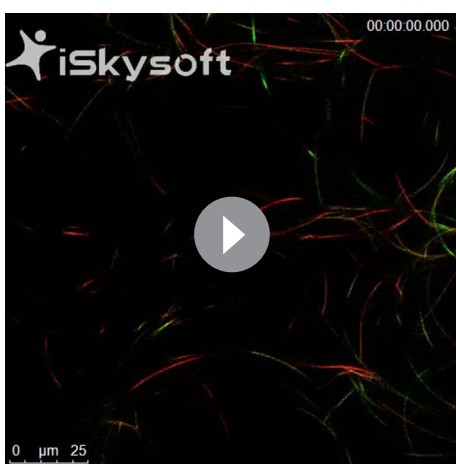

**Video 3.** Sperm motility assays in w1118/Oregon R males ('nuclear suppressor' background) carrying COIIG177S mutant mtDNA. From a representative male grown at 29°C. Sperm stained with mitotracker Green (green), which stains immotile sperm and mitotracker CMS Rox (red), which stains motile sperm preferentially (related to *Figure 10*).

## Materials and methods

### *D. melanogaster* maintenance and strains

All *D. melanogaster* strains were maintained on standard molasses-cornmeal medium at 25°C. We obtained the w1118 strain of *D. melanogaster* free of Wolbachia contamination (kind gift of Susan Parkhurst, FHCRC). We obtained the *D. melanogaster* DGRP lines from the Bloomington Drosophila Stock Center (Bloomington, IN). The other *D. melanogaster* lines were either obtained from the Drosophila Species Stock Center (San Diego, CA) or from our colleagues (Daven Presgraves, John Pool, Chip Aquadro).

We confirmed absence of Wolbachia using PCR in the original w1118 stock. Briefly, six flies from each of the assayed strains were homogenized in 49 µl squish buffer (10 mM Tris-CL pH 8, 1 mM EDTA, 25 mM NaCl) and 1 µl Proteinase K (200 µg/ml), following which the samples were incubated at 37°C for 30 min, heated to 95°C for 10 min and cooled to room temperature for 10 min. We then used WSP (Wolbachia surface protein) primers (5'- GCATTTGGTTAYAAAATGGACGA-3' and 5'-GGAGTGATAGGCATATCTTCAAT-3') and performed PCR using Taq DNA polymerase (New England Biolabs, Ipswich, MA). We also assessed the status of Wolbachia infection after 35 generations of experimental evolution,in the re-isolated wt and COII$^{G177S}$ mtDNA lines (*Figure 1—figure supplement 1*). All samples tested were found to be negative for Wolbachia.

### Experimental evolution

An original stock was established by expanding the w1118 strain from one bottle to 36 bottles. We established 12 experimental lines and 6 coevolving lines in bottles by adding 300 virgin females and 100 males from the original stock. The bottles were flipped three times a day for two days after which the flies were discarded. To set up every subsequent generation, virgin females were collected over a course of seven days and then crossed with males from the original stock in the case of the experimental lines. For coevolving lines, males from the corresponding coevolving line were used. After 35 generations, all lines were maintained by flipping bottles allowing sibling mating.

### Fertility and sex ratio assays

All fertility and sex ratio assays were done in vials with five virgin females and three virgin males that were all 2–5 days old. All assays were done in 10 replicates per group. Flies were transferred to new vials every 2–3 days and flies were discarded after the five$^{th'}$ 'flip'. In order to assess overall fertility, we counted all adult progeny to emerge from all of the vials.

### Whole genome sequencing

100 females from the original stock and experimental line 7 were lysed in 100 µl squish buffer (10 mM Tris-HCl pH 8, 1 mM EDTA, 25 mM NaCl, 200 µg/ml Proteinase K) with RNase A at final concentration of 30ng/ml. Lysate was incubated at 55°C for 1 hr followed by 95°C for 10 min. Total DNA was then phenol-chloroform extracted from the lysates and the DNA pellet was dissolved in 25 µl H$_2$O. Library preparation of samples and paired-end sequencing was performed by the Genomics Shared Resource core at Fred Hutchinson Cancer Research Center on an Illumina HiSeq 2500. Sequence data are available through NCBI's SRA database (project SRP057279).

Whole genome sequencing data was analyzed by aligning reads to a modified version of the *D. melanogaster* reference genome assembly (BDGP Release 5), where we masked a region of the unassembled chromosome (chrU:5288528–5305749) that harbors an alternative version of the mitochondrial genome sequence (in order to prevent ambiguous mapping of mitochondrial reads to the chrU region). Reads were filtered, quality- and adapter-trimmed, and aligned to our custom reference genome using GSNAP (*Wu and Nacu, 2010*). SNPs were called in the non-repetitive region of the mitochondrial genome (chrM:1–14196) using GATK's HaplotypeCaller algorithm (*DePristo et al., 2011*) with a 'sample_ploidy' setting of 20. A G-to-A change at position chrM:3611, encoding COII$^{G177S}$ (*Figure 3*), was present in 98% of Experimental Line 7 reads and 59% of reads from the original stock. No other variants had greater than 5% higher read frequency in Experimental Line 7 than in the original w1118 sample.

## Duplex sequencing

Duplex Sequencing of fly mtDNA was performed as previously described (*Kennedy et al., 2014*) with several modifications. Total DNA was purified from individual fly heads using a QIAamp DNA micro kit (Qiagen Inc., Germantown MD). The DNA was sonicated using a Covaris AFA S2 ultrasonicator (Covaris Inc., Woburn MA) with the following settings: Duty cycle: 10%; Intensity: 5; Cycles/burst: 100; Time: 15 s × 3. The sheared DNA was then end-repaired and ligated using the NEBNext Ultra 2 end-repair and dA-tailing kit (New England Biolabs, Ipswich MA) according to the manufacturer's instructions. Duplex Sequencing adapters, described previously (*Kennedy et al., 2014*), were ligated to the DNA library using the NEBNext Ultra 2 ligation kit (New England Biolabs, Ipswich MA) according to the manufacturer's instructions. 1.5 ng of total DNA was then PCR amplified using KAPA HiFi DNA polymerase (Roche Inc., Basel Switzerland) according to the manufacturer's recommendations. After amplification, the mtDNA was enriched by targeted capture using xGen target capture probes (Integrated DNA Technologies Inc., Coralville IA) specific to the fly mitochondrial genome. The samples were then sequenced on a NextSeq500 machine (Illumina Inc., San Diego, CA) to generate 150 bp paired-end reads. The data were processed as previously described (*Kennedy et al., 2014*).

## Establishing wildtype and COII$^{G177S}$ lines from the original stock

15 virgin females were collected from the original w1118 stock. Each female was individually mated with males from the same stock. Status of the G177S allele was assessed by Sanger sequencing of the females after they were allowed to have progeny. One line that appeared to be homoplasmic wildtype and one line that appeared to be homoplasmic mutant were kept and constituted the re-established stocks.

## Cox activity

six females or eight males from each group were gently homogenized on ice in 50 µl of sodium phosphate buffer with 0.05% Tween-20. Lysates were centrifuged at 4°C at 4000 × g for 1 min. Supernatant was collected and 20 µl were used to measure COX activity using a kit (ScienCell, Carlsbad, CA). COX activity was normalized to total protein concentration as determined by Pierce BCA Protein Assay Kit (Pierce Biotechnology, Rockford, IL). Data shown represent averages of 4–5 replicates per group.

## ROS levels

10 flies 3–4 days old and raised at 29°C were homogenized on ice in 200 µl PBS with 0.1% Tween-20. Lysates were centrifuged at 4°C at 13,000 × g for 10 min. The supernatant was collected and 100 µl was incubated with 50 µM H2DCF (Molecular Probes). Fluorescence intensity was measured in an Infinite M1000Pro (Tecan, Switzerland) microplate reader using 490 nm wavelength excitation and 520 nm wavelength emission. ROS levels were normalized to total protein concentration as determined by Pierce BCA Protein Assay Kit. Data shown represent averages of eight replicates per group.

## ATP levels

Five males 3–4 days old and raised at 29°C were homogenized on ice in 100 µl guanidine extraction buffer (6 M guanidine HCl, 100 mM Tris-HCl, pH 7.3). Samples were frozen in liquid nitrogen for 5 min and then incubated at 95°C for 5 min. Lysates were centrifuged at 4°C for 10 min at 12,000 × g. Supernatant was collected and 5 µl was diluted in 95 µl H$_2$O. 10 µl of the diluted lysate was used to measure ATP levels using the ATP determination kit (Cat #: A22066, Molecular Probes). ATP levels were normalized to total protein concentration as determined by Pierce BCA Protein Assay Kit. Data shown represent averages of four replicates per group.

## Mating-induced egg laying

Five virgin wildtype females were either kept alone, or mated with three wildtype or COII$^{G177S}$ mutant males at 29°C in vials with grape plates. Flies were flipped into new vials every day for eight days. 24 hr after the flies were removed from a vial, number of unhatched eggs and larvae on the grape plate were counted. Data shown represents averages of six replicates per group.

## Testis imaging

Immunofluorescence staining of testes was performed as described previously (*Cheng et al., 2008* ). Briefly, testes were dissected in PBS, transferred to 4% formaldehyde in PBS and fixed for 30–60 min. The testes were then washed in PBS-T (PBS containing 0.1% Triton-X) for at least 30 min and mounted in VECTASHIELD with DAPI (Vector Labs). Imaging of whole testis was performed on a Leica SP8 confocal microscope.

## Sperm imaging

Unfixed seminal vesicles were dissected in PBS mounted onto slides with PBS containing 1 µM Mitotracker Green and Mitotracker CMXRos (Lifetechnologies). Sperm were extruded from seminal vesicles using a tungsten needle. Imaging was performed immediately upon addition of a cover slip to minimize the effects of hypoxia. Still images were taken on a Leica SP8 confocal microscope and movies were captured using a resonant scanner on a Leica SP5 confocal microscope.

## Suppressor assays

Males from wild strains including the DGRP collection were crossed with w1118 females homoplasmic for either wildtype or COII$^{G177S}$ mtDNA. Resulting heterozygous male progeny were assayed for fertility, sperm development and motility, and COX activity.

## Lifespan assay

An assay for lifespan was performed as previously described (*Burman et al., 2014*). Briefly, 100 flies from each group were divided into five vials with 20 flies each. Vials were flipped every other day and the number of dead flies was counted until all the flies had died.

## Bang sensitivity

An assay for bang sensitivity was performed as previously described (*Burman et al., 2014*). Briefly, vials with two flies each were mechanically stimulated by placement in a bench-top vortex for 10 s at the maximum setting. The time for each fly to right itself after vortexing was recorded. Data shown represents average from 18–20 flies per group.

## Heat tolerance

Assay for resistance to heat-induced paralysis was performed as previously described (*Burman et al., 2014*). Briefly, flies were assayed for heat-induced paralysis by placing them into pre-warmed vials maintained at 39°C. The time for flies to become paralyzed was recorded. After exposure to 39°C for 6 min the animals were then placed in new room-temperature vials (~20°C), and the recovery time from paralysis was recorded. Data shown represents average from 16 flies per group.

# Acknowledgements

We thank Lisa Kursel, Tera Levin, Mia Levine, Nitin Phadnis, and Benjamin Ross for comments on the manuscript. The authors were supported by a postdoctoral fellowship from the Helen Hay Whitney Foundation (MRP), the Mathers Foundation (HSM), and US National Institute of Health grants F30 AG045021 (HY), GM104990 (LP) and GM074108 (HSM). YMY and HSM are Investigators of the Howard Hughes Medical Institute.

# Additional information

### Competing interests

YMY: Reviewing editor, *eLife*. The other authors declare that no competing interests exist.

### Funding

| Funder | Grant reference number | Author |
| --- | --- | --- |
| Helen Hay Whitney Foundation | | Maulik R Patel |

| National Institutes of Health | F30AG045021 | Heiko Yang |
| Howard Hughes Medical Institute | | Yukiko M Yamashita<br>Harmit S Malik |
| National Institute of General Medical Sciences | GM104990 | Leo J Pallanck |
| G Harold and Leila Y. Mathers Foundation | | Harmit S Malik |
| National Institute of General Medical Sciences | GM074108 | Harmit S Malik |

The funders had no role in study design, data collection and interpretation, or the decision to submit the work for publication.

## Author contributions

MRP, LJP, Conception and design, Acquisition of data, Analysis and interpretation of data, Drafting or revising the article; GKM, Acquisition of data, Analysis and interpretation of data, Drafting or revising the article; AJL, KT, SRK, Acquisition of data, Analysis and interpretation of data; HY, Acquisition of data, Analysis and interpretation of data, Contributed unpublished essential data or reagents; JMY, Analysis and interpretation of data, Drafting or revising the article; YMY, Acquisition of data, Analysis and interpretation of data, Drafting or revising the article, Contributed unpublished essential data or reagents; HSM, Conception and design, Analysis and interpretation of data, Drafting or revising the article

## Author ORCIDs

Janet M Young, http://orcid.org/0000-0001-8220-8427
Harmit S Malik, http://orcid.org/0000-0001-6005-0016

# Additional files

## Major datasets

The following dataset was generated:

| Author(s) | Year | Dataset title | Dataset URL | Database, license, and accessibility information |
|---|---|---|---|---|
| Maulik R Patel, Ganesh K Miriyala, Aimee J Littleton, Heiko Yang, Kien Trinh, Janet M Young, Scott R Kennedy, Yukiko M Yamashita, Leo J Pallanck, Harmit S Malik | 2015 | A mitochondrial DNA hypomorph of cytochrome oxidase specifically impairs male fertility in Drosophila melanogaster | http://trace.ncbi.nlm.nih.gov/Traces/sra/?study=SRP057279 | Publicly available at NCBI Sequence Read Archive (accession no: SRP057279) |

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
