## [Decision Letter]

[Editors’ note: a previous version of this study was rejected after peer review, but the authors submitted for reconsideration. The first decision letter after peer review is shown below.]

Thank you for choosing to send your work entitled "Experimental evolution uncovers a male-harming mitochondrial mutation in *Drosophila melanogaster*" for consideration at *eLife*. Your full submission has been evaluated by K VijayRaghavan (Senior editor), a Reviewing Editor and two peer reviewers and the decision was reached after discussions between the reviewers. Based on our discussions and the individual reviews below, we regret to inform you that your work will not be considered further for publication in *eLife*.

As you will see from the reviewers' comments below, one of the primary concerns was that your claim of establishing a novel experimental evolution scheme to isolate male-harming mitochondrial DNA mutations is not supported by data. Without more compelling evidence for this scheme, the current manuscript is not suitable for *eLife*. In addition, the link between the genotype of the missense mitochondrial mutation (COII^G177S^) and phenotype described is complicated by the fact that the phenotype requires high temperature. Finally, in light of these concerns, the effect of nuclear modifiers was of great interest, but it would require further development, such as identification and characterization, to elevate the manuscript.

Reviewer #1:

One major concern is the claim that the experimental model was "successful" in isolating the male-specific phenotype. As the allele already existed in the parental stock, it may simple be that the authors were able to – by chance alone – fix the allele in 1-in-18 lines, and not due to the breeding scheme. Some added detail as to the state of the allele in the other lines would help us to determine if the breeding scheme can actually be said to be successful in driving the emergence of this phenotype, or if the experiment was simply very lucky to catch this one allele. Some additional arguments in support of their method would be very helpful.

The allele in question appears to have been present in the parental stock, and was isolated to homoplasmy in 1 of the 18 experimental lines. Is it possible for the authors to test the dynamics of the heteroplasmic transmission in the other 17 lines? While the authors are unable to detect any defects aside from the male fertility, if the transmission of the allele is consistent with neutral drift, the authors would clearly demonstrate that the allele had no fitness effects until fixed in the population and would very strongly support the claim of a specific phenotype. However, evidence of negative selection would show the some difficult-to-detect deficiency in the flies with the allele, and a positive selection signal would set up an interesting scenario of the allele being beneficial to the population, but detrimental to male fertility when fixed.

Reviewer #2:

This manuscript addresses an important issue in biology and describes a potentially interesting male semi-sterile mutation. The reported results can be divided into three areas. In the first, the manuscript describes a strategy described as "a novel experimental evolution scheme in *Drosophila melanogaster* to recover male-harming mutations". In the second, the phenotype of the temperature-enhanced sterility of the one mutation recovered is characterized. In the third, modification of the sterility affects by nuclear background is described. As discussed below, I found that the first part to harbor significant potential interest, but felt that the paper did not support the efficacy of the practice that it advocated. Nonetheless, they retrieved a male harming mutant, and second part of the manuscript describes its characterization. While this is an interesting mutant that fits the bill of a male harming mutant, I was somewhat uncertain about the value of the details of the phenotypic characterization. It seemed to me the details of the phenotype are likely to depend on the use of a high temperature to increase the expressivity of the defect. The analysis of enhancers in the third part of the paper is of great interest, but it was not advanced to level that gives us any insight into the nature of the interactions. In this regard, I was again concerned about the use of high temperature to enhance the phenotype. No attempt was made to disambiguate the influence of genetic background effects on the temperature range of fertility from a specific effect on the mutation, leaving a nagging suspicion that the effect of the nuclear background might not be very specific.

The strategy that was presented as a method to allow accumulation of male-harming mutations without hypothetical secondary counter selection due to male sterility was to simply out cross the females to an independent population of males. There are several issues with the claimed importance of this strategy and questions regarding its effectiveness that the manuscript fails to address.

One tangential, but in my mind important, consideration is that outcrossing may be an experimental condition in the laboratory but it is standard in the wild. If the rationale were correct, the inbreeding of confined laboratory stocks should be protecting them from the emergence of male harming mutants. In this regard, it would be interesting to compare the frequency of male harming mtDNA mutations in well-mixed wild populations, to the frequency in the confined breeding populations of laboratory flies. According to the rationale, one might expect a substantial frequency of male harming mutants in wild populations that then fail when established as more inbred laboratory stocks.

But the most important question is whether the paper shows that the strategy was effective. I was not convinced that it did. The only recovered mutation was in the starting population at a very high level. The paper suggests in its wording that this was a fleeting thing caught by the breeding strategy: e.g. It is suggested that "the mutant allele likely arose de novo in the original stock and transiently existed in a state of heteroplasmy before approaching homoplasmy in EL7", and "we devised an experimental evolution scheme to absolve mtDNA of supporting male function while both maintaining selection on female mtDNA function as well as reducing the possibility of nuclear suppression or male-harming mtDNA. Our scheme uncovered a COII^G177S^ mtDNA mutation[…]". The idea of fleeting mutation present in the original stock that was captured by the "experimental evolution scheme" might support the idea that it was effective. However, it seems very unlikely that the mutation was fleeting and there is no evidence that the scheme was needed to capture it. Even rare changes can go to fixation in small populations and one out six assayed flies in the starting population was homoplasmic for the mutation. Over 35 generations (plus an unspecified number of generations of additional crosses in which the flies were allowed to self) drift will lead to dramatic fluctuations in the abundance of the homoplasmic line and to random fixation. Fixation of an already abundant mutation in one of 18 lines is no surprise and there is no foundation for a conclusion that this possibility was enhanced by the "experimental evolution scheme".

The only data mentioned to support the idea that the mutation was present transiently in the original stock were not shown and the quoted finding does not seem directly relevant: "heteroplasmic flies were eventually lost over the course of a year from the original stock after their initial detection; later analyses revealed the population now only consists of individuals carrying homoplasmic wildtype or mutant mtDNA." The loss of the heteroplasmic flies is only indirectly relevant. Since the mutant persisted in the population as homoplasmic flies there is no indication that its presence was transient. Additionally, based on Solignac's (1984) detailed description of the gradual and statistical segregation of mtDNA genotypes one can make inferences about the behavior of a mutation in a stock – whether it is drifting randomly or is affected by selection. The only data shown regarding the distribution of heteroplasmy was an analysis of six flies of the original population. This is woefully inadequate for any detailed consideration of levels of heteroplasmy. A more thorough analysis might have revealed whether there was any constraint limiting the propagation of the mutant allele in the original stock, since such constraints, by selecting for different progeny classes, will alter the unbiased statistical changes in the distribution of heteroplasmy levels. Demonstration that a constraint existed would provide some foundation for the proposal that the "experimental evolutionary scheme" would have consequence-relief of this constraint. The crosses presented in this manuscript could have simply provided opportunity for random drift to fix an abundant pre-existing mutation. To make the claims regarding the "experimental evolutionary scheme", the manuscript would have to dismiss this null hypothesis and demonstrate the efficacy of the approach. At present, the appearance of one line out 12 with a mutant in the experimental sample and none in the 6 control lines is far from enough to argue that the outcrossing was essential to the recovery of the mutation.

The “experimental evolutionary scheme" was designed to bypass the influence of indirect counter selection due to reduced fertility of the males: but to what extent does such a selection influence outcome? It appears that the heteroplasmic male flies are fertile (not tested, but failure to see fertility depression in lines with high levels of mutant genomes suggest that mutant genomes can be carried without a phenotype) and hence that no such counter selection would operate early after the appearance of the mutant allele (when it is rare and heteroplasmic). The persistence of mutant flies (perhaps heteroplasmic, but untested) in three of the six coevolving/control lines suggests the absence of a significant counter selection preventing their maintenance. Consequently, male sterility could only come into play, if at all, very late after the mutation rose to be a dominant allele with frequent production of homoplasmic derivatives, and since the male sterility phenotype is weak at the temperatures used for propagation, it would not be surprising that secondary selection due to male fertility could prevent fixation of the mutant. Furthermore, since the scheme did not use unrelated males for outcrosses but used males from the starting population, which apparently harbored the mutant for many generations (many generations are essential to produce homoplasmic flies), these males might already have carried suppressors of the phenotype. Thus, I do not think that manuscript offers support for claim that an effective scheme for recovery of male-harming mutations is described. I would like to see direct evidence that it is effective and efficient.

The second component of the manuscript, the description of the phenotype of the male sterile mutation is detailed but I was not impressed that it gave us much insight into the defects and their connection to the mutation. One obfuscating issue is the use of high temperature to enhance the phenotype of an otherwise weak expressivity of the mitochondrial mutation. The highest temperature at which flies can be cultivated is generally limited by male sterility (e.g. David JR et al., 2005). For laboratory stocks, this highest temperature is usually taken as 29°C but populations tend to show differences in the temperature range across which they will grow-perhaps not surprisingly as environmental variables and adaptation would select for different ranges. At the temperature limit, defects in sperm development are usually seen as responsible for sterility (e.g. Rohmer C, et al., 2004). Interactions between nucleus, mitochondria and temperature have been described as affecting male sterility (Hoekstra et al., 2013). The manuscript shows an interaction of temperature and the mitochondrial mutation, but the study of the phenotype leaves an uncertainty whether the phenotype described is primarily that of the mutant or that produced at high temperature. That is, one might imagine that the mutation might sensitize the flies so that the high temperature defect shows up at 29°C, or one can imagine that 29°C increases the expressivity of the mutant phenotype, or something in between. In this regard, it would be nice to know if the sterile phenotype of *w1118* wild type flies when raised at temperatures beyond 29°C, resembles or is distinct from the phenotype as seen in the mutant at 29°C.

The analysis of nuclear interactions is potentially exciting, but the reported analysis represents a first step. The loci responsible are not mapped or otherwise identified. Sequenced strains are analyzed but no mention is made of any finding emanating from this. I worry that modification might be the result of modification of temperature sensitivity. The things underlying this concern are that the maximal temperature of male fertility commonly vary between strains, genetic backgrounds that modified the reported phenotype are remarkably common, and the data shown for 29°C in the supplement of Figure 9 shows that *w1118* is unusually weak even at the temperature selected as widely permissive. If the temperature is raised to say 31°C, the wild type *w1118* stock is likely to be male sterile: I would like to know whether temperature tolerance is increased (i.e. this male sterility expected for the wild-type *w1118* strain is suppressed) by out crossing of the wild type to the same strains that suppressed the mutant. This control might reveal that the recorded affects are due to expected widespread differences in thermo-tolerance. This concern that the modification may be due to a more generic affect is partly alleviated by the reported increase in Cox activity in suppressed mutant backgrounds.

[Editors’ note: what now follows is the decision letter after the authors submitted for further consideration.]

Thank you for submitting your article "A mitochondrial DNA hypomorph of cytochrome oxidase specifically impairs male fertility in *Drosophila melanogaster*" for consideration by *eLife*. Your article has been reviewed by two peer reviewers, and the evaluation has been overseen by a Reviewing Editor and K VijayRaghavan as the Senior Editor. The reviewers have opted to remain anonymous.

The reviewers have discussed the reviews with one another and the Reviewing Editor has drafted this decision to help you prepare a revised submission.

Summary:

In their final submission the authors should address better previous criticism influenced by the fact the supposed approach to isolate male harming mutants was the most innovative part of the original submission. The new version is hence weaker, but still a nice report and should be published at *eLife*. Of two previous criticisms, the bigger one is that the data are not sufficient to document homoplasmy. While this could be serious flaw, it is very likely that the line is close to homogeneous and if it were not and the authors sampled flies that were heteroplastic in their tests of fertility they would have seen phenotypic variation. If the authors answer in this way, and put more guarded statements in their discussion, we would be satisfied on this account. The are other issues, on the phenotypic analysis which are more about making things more clear, particularly aspects that are buried in the supplementary data. Eliminating some unfounded discussion will bring more sharpness, but this is something that the authors are best placed to address. These are detailed below.

Essential revisions:

The presentation of the data has been changed substantially, while the data remain much the same. The change in the presentation has resolved one of the central concerns about the previous version of the manuscript that presented a crossing scheme as an effective selection for male harming mutations. The critiques, certainly mine, argued that this was not supported by the data and that the mitochondrial mutation preexisted and was uncovered by chance. The present draft still argues that this scheme should be effective, which I severely doubt based on pragmatics of mutation frequency and infrequent segregation of homoplasmic lines. I think that the unsubstantiated sentence in the current version saying that "we reasoned that multiple mutations per mtDNA would be sampled in 35 generations across 18 lines" should be eliminated or justified.This is completely untenable as shown by the mathematical analysis of the number of generations required for lines to progress to homoplasmy (See Solignac). Nonetheless, the major change in presentation has corrected the biggest problem with the previous submission.

The question now is, what is the merit of the present manuscript. The authors do report a maternally inherited partial male sterility trait that is linked with an SNP mt:CoII. The recognized SNP may well be the cause of the phenotype, but the possibility of linked changes in the unsequenced 4kb regulatory region was not eliminated (note that this is the region that changes the most rapidly and spontaneous variants of this region have been shown to arise in laboratory stocks – see Rand). Furthermore, identification of a mitochondrial DNA mutation altering an electron transport component that causes male sterility is not entirely novel (see Xu). The manuscript does report a fairly detailed effort to characterize the defect and shows suppression of this defect in crosses that introduce different nuclear backgrounds. In this regard, it goes beyond other reports that argued for nuclear modification of phenotypes caused by mitochondrial mutations. In our opinion these findings are of interest and could constitute a significant advance. We are however, left with some hesitancy because the merit ought to lie in the rigor in of the documentation of the principles, and we see some shortcomings.

One concern is that the data are not sufficient to rigorously demonstrate the genetic homogeneity of the lines examined. The details are below, but this view can be summarized as follows: the original plan treated the crosses as if they were a selection that would yield a pure male harming mutation, and suitable measures where not take to ensure the genetic homogeneity of the stock that was retrieved. Given the number of mt genomes carried by flies, a high sensitivity of detection of a rare allele (in a range approaching 0.1%) and analysis of single flies is needed to rigorously demonstrate homoplasmy.

A second issue concerns data on fertility demonstrating that in an early first mating the males exhibited no (at 25°C) or weak (29°C) deficits in fertility. At least in reading the text, the subsequent analyses did not seem to take this fully into account. Many of my concerns here could be dealt with by better reference to supplementary material and more careful wording. Nonetheless, it still seemed to me that this issue was not fully investigated and the impact of this finding on other analyses was not always considered.

Overall, this an interesting report that suffers from weakness in its details. These can be addressed in the revised final submission

---

## [Author Response]

[Editors’ note: the author responses to the first round of peer review follow.]

*As you will see from the reviewers' comments below, one of the primary concerns was that your claim of establishing a novel experimental evolution scheme to isolate male-harming mitochondrial DNA mutations is not supported by data.*

We completely agree with this criticism. While we obtained the COII^G177S^ mtDNA mutation via our experimental evolution scheme, our subsequent experiments were able to determine that this was a pre-existing heteroplasmic mtDNA mutation that manifested its male infertility only upon fixation in one of the experimental lines (el7). For the purposes of the current manuscript, we now describe the experimental evolution scheme simply to detail how we found a specifically male-harming mtDNA mutation. We also acknowledge that this mutation might have been obtainable without the scheme with a priori knowledge of the heteroplasmy (which we did not have), and that significantly more work is needed to determine whether the scheme will be effective. We have accordingly significantly edited our manuscript, including changing the title.

*In addition, the link between the genotype of the missense mitochondrial mutation (COII^G177S^) and phenotype described is complicated by the fact that the phenotype requires high temperature.*

One of the reviewers raised the concern that the male fertility defects caused by the COII^G177S^ mutation might be confounded by the use of high temperature. We performed mutant characterization in the original submission at high temperature because the fertility phenotype is more pronounced in young males at high temperature. However, we agree with the reviewer concerns and have now performed additional experiments to address these points. We also showed in the original submission that the fertility phenotype is also present at 25°C but it manifests in older males. We now show that the cellular phenotype of the COII^G177S^ mtDNA mutation is similar in older males at 25°C to the phenotype in young males at 29°C. We further show that sterility of young males at 29°C is suppressed by a nuclear background (OregonR) in a manner similar to the suppression of old males at 25°C. Thus, we feel that we have completely addressed with our new experiments the reviewer concerns that our sterility phenotype and its suppression are confounded by a requirement of high temperature. It bears reiterating that we performed many life-history trait measurements at both 25°C and 29°C; male fertility was the only phenotype affected suggesting that the sterility defect associated with COII^G177S^ mtDNA is highly specific.

Finally, in light of these concerns, the effect of nuclear modifiers was of great interest, but it would require further development, such as identification and characterization, to elevate the manuscript.

There are two points to be made here. First, the abundance of nuclear suppressor(s) in *D. melanogaster* directly bears out the predictions of the theory of mito-nuclear genetic conflict and reinforces our finding that COII^G177S^ is a male-harming mtDNA mutation.. Second, we would love to know the genetic identity of the suppressor. But the reviewer is doubtless aware that this is a significant endeavor involving at least a few years of work. The best demonstration of this suppressor mapping was a recent paper (Meikeljohn et al. PLOS Genetics 2013), which found a nuclear-encoded tRNA synthetase that epistatically interacts with and suppresses a mitochondrial tRNA mutation. This work appeared three years after the original observation of the incompatibility (Montooth et al. Evolution 2010). Furthermore, the identity of this suppressor mutation was suggested by the very nature of the mtDNA mutation in a tRNA gene. Our preliminary experiments and the fact that we see both partial and completely restoration of fertility suggest that the suppression involves multiple genetic loci, whose identification will require 2-3 years of additional work. We intend to aggressively pursue this, but we feel that lack of this identification does not affect the novelty or importance of our findings in the current study.

Isolation of a bona fide male-harming mtDNA mutation

Although this is not a point explicitly mentioned by the editor, we would like to reinforce our claim that the identification of a specifically male-harming mtDNA mutation is novel and to our knowledge unprecedented in animals. The closest that any studies have come to characterization of ‘male-harming’ mtDNA variants are the elegant series of experiments performed by the group of Damian Dowling, who showed that mtDNA haplotypes from different *D. melanogaster* strains affect aging in males but not females (Camus et al. 2012 Current Biology), and competitive male fertility (Yee et al. 2013 Current Biology). These studies did not comprehensively test the role of all mtDNA haplotypes on female fitness. Just to highlight the novelty of our finding, while our paper was under review at *eLife*, one mtDNA haplotype that was previously shown to deleteriously affect competitive male fertility was also shown to decrease female lifespan (Camus et al., 2015 Current Biology); such a mutation could not be considered specifically ‘male-harming’. In contrast, our work examined the impact of COII^G177S^ mutation on both males and females in the context of fertility, lifespan, neurological function, and stress resistance, each at normal and elevated temperatures, and in young and old individuals. Hence, we can conclude with significantly greater confidence than any other study published so far that a single mtDNA variant in metazoans can be specifically detrimental to males but neutral in females. We believe that this comprehensive characterization of the COII^G177S^ mutation is the most novel and impactful finding of our manuscript. We have now revised our manuscript to emphasize this point.

We hope the reviewers will agree that the revised manuscript addresses all of their concerns, most of which arise from our initial claim regarding the ‘success’ of the experimental evolution strategy. We further hope the reviewers will agree that our demonstration of COII^G177S^ as a specifically male-harming mtDNA mutation in animals is highly significant and impactful.

*Reviewer #1:*

One major concern is the claim that the experimental model was "successful" in isolating the male-specific phenotype. As the allele already existed in the parental stock, it may simple be that the authors were able to – by chance alone – fix the allele in 1-in-18 lines, and not due to the breeding scheme. Some added detail as to the state of the allele in the other lines would help us to determine if the breeding scheme can actually be said to be successful in driving the emergence of this phenotype, or if the experiment was simply very lucky to catch this one allele. Some additional arguments in support of their method would be very helpful.

As discussed earlier, we agree with the reviewer. We now acknowledge that although we obtained the COII^G177S^ mutation via the experimental evolution scheme, it may have been possible to fix this allele to homoplasmy without our breeding scheme. We have significantly revised the beginning of the manuscript to reflect this acknowledgement, and to emphasize our recovery of the COII^G177S^ mutation, which appears to be specifically detrimental to males and is the most significant finding of our work.

The allele in question appears to have been present in the parental stock, and was isolated to homoplasmy in 1 of the 18 experimental lines. Is it possible for the authors to test the dynamics of the heteroplasmic transmission in the other 17 lines? While the authors are unable to detect any defects aside from the male fertility, if the transmission of the allele is consistent with neutral drift, the authors would clearly demonstrate that the allele had no fitness effects until fixed in the population and would very strongly support the claim of a specific phenotype. However, evidence of negative selection would show the some difficult-to-detect deficiency in the flies with the allele, and a positive selection signal would set up an interesting scenario of the allele being beneficial to the population, but detrimental to male fertility when fixed.

We do not have longitudinal data from all our experimental and coevolved lines, but the data we do have suggests that the fixation of the COII^G177S^ mutation did not occur until at least 25 generations through our experiment. Since our experimental lines would be expected to be homogeneous at the nuclear level given our backcrossing scheme, our findings that the viability and fertility of the EL lines (except EL7) are like wildtype despite being highly variable for the COII^G177S^ heteroplasmic mutation lead us to favor the conclusion that the mutation is neutral, not beneficial, to the population, but is detrimental to male fertility on homoplasmy.

*Reviewer #2:*

*This manuscript addresses an important issue in biology and describes a potentially interesting male semi-sterile mutation. The reported results can be divided into three areas. In the first, the manuscript describes a strategy described as "a novel experimental evolution scheme in Drosophila melanogaster to recover male-harming mutations". In the second, the phenotype of the temperature-enhanced sterility of the one mutation recovered is characterized. In the third, modification of the sterility affects by nuclear background is described. As discussed below, I found that the first part to harbor significant potential interest, but felt that the paper did not support the efficacy of the practice that it advocated. Nonetheless, they retrieved a male harming mutant, and second part of the manuscript describes its characterization. While this is an interesting mutant that fits the bill of a male harming mutant, I was somewhat uncertain about the value of the details of the phenotypic characterization. It seemed to me the details of the phenotype are likely to depend on the use of a high temperature to increase the expressivity of the defect. The analysis of enhancers in the third part of the paper is of great interest, but it was not advanced to level that gives us any insight into the nature of the interactions. In this regard, I was again concerned about the use of high temperature to enhance the phenotype. No attempt was made to disambiguate the influence of genetic background effects on the temperature range of fertility from a specific effect on the mutation, leaving a nagging suspicion that the effect of the nuclear background might not be very specific.*

*The strategy that was presented as a method to allow accumulation of male-harming mutations without hypothetical secondary counter selection due to male sterility was to simply out cross the females to an independent population of males. There are several issues with the claimed importance of this strategy and questions regarding its effectiveness that the manuscript fails to address.*

*One tangential, but in my mind important, consideration is that outcrossing may be an experimental condition in the laboratory but it is standard in the wild. If the rationale were correct, the inbreeding of confined laboratory stocks should be protecting them from the emergence of male harming mutants. In this regard, it would be interesting to compare the frequency of male harming mtDNA mutations in well-mixed wild populations, to the frequency in the confined breeding populations of laboratory flies. According to the rationale, one might expect a substantial frequency of male harming mutants in wild populations that then fail when established as more inbred laboratory stocks.*

This is an interesting suggestion and indeed, the main topic of a paper (Wade & Brandvain 2008 *Evolution*) that predicts exactly what the reviewer suggests – i.e. inbreeding selects against male-harming mtDNA mutations due to indirect selection. Indeed, this theoretical claim served as the rationale for our experimental evolution strategy. We took the predictions from this theoretical study and used it to motivate our scheme to isolate male-harming mtDNA mutations. The reviewer makes an interesting suggestion that we plan to follow up on by making use of the DGRP collection in the future. However, that work would be a significant undertaking that is beyond the scope of this manuscript.

*But the most important question is whether the paper shows that the strategy was effective. I was not convinced that it did. The only recovered mutation was in the starting population at a very high level. The paper suggests in its wording that this was a fleeting thing caught by the breeding strategy: e.g. It is suggested that "the mutant allele likely arose* de novo *in the original stock and transiently existed in a state of heteroplasmy before approaching homoplasmy in EL7", and "we devised an experimental evolution scheme to absolve mtDNA of supporting male function while both maintaining selection on female mtDNA function as well as reducing the possibility of nuclear suppression or male-harming mtDNA. Our scheme uncovered a COII^G177S^ mtDNA mutation[…]". The idea of fleeting mutation present in the original stock that was captured by the "experimental evolution scheme" might support the idea that it was effective. However, it seems very unlikely that the mutation was fleeting and there is no evidence that the scheme was needed to capture it. Even rare changes can go to fixation in small populations and one out six assayed flies in the starting population was homoplasmic for the mutation. Over 35 generations (plus an unspecified number of generations of additional crosses in which the flies were allowed to self) drift will lead to dramatic fluctuations in the abundance of the homoplasmic line and to random fixation. Fixation of an already abundant mutation in one of 18 lines is no surprise and there is no foundation for a conclusion that this possibility was enhanced by the "experimental evolution scheme".*

As we have responded to (points 1 and 5 above), we agree with the reviewer’s concern. We have significantly revised the manuscript to remove any reference to the ‘success’ of our experimental evolution strategy but simply to state that this is how we obtained the COII^G177S^ mutation. We remain confident that the mutation we did obtain is specifically male-harming and therefore its characterization important and, to our knowledge, unprecedented. As the reviewers point out, the effectiveness of the experimental evolution approach has yet to be rigorously tested. A full proof-of-principal test of the strategy will be the focus of future work.

The only data mentioned to support the idea that the mutation was present transiently in the original stock were not shown and the quoted finding does not seem directly relevant: "heteroplasmic flies were eventually lost over the course of a year from the original stock after their initial detection; later analyses revealed the population now only consists of individuals carrying homoplasmic wildtype or mutant mtDNA." The loss of the heteroplasmic flies is only indirectly relevant. Since the mutant persisted in the population as homoplasmic flies there is no indication that its presence was transient. Additionally, based on Solignac's (1984) detailed description of the gradual and statistical segregation of mtDNA genotypes one can make inferences about the behavior of a mutation in a stock – whether it is drifting randomly or is affected by selection. The only data shown regarding the distribution of heteroplasmy was an analysis of six flies of the original population. This is woefully inadequate for any detailed consideration of levels of heteroplasmy. A more thorough analysis might have revealed whether there was any constraint limiting the propagation of the mutant allele in the original stock, since such constraints, by selecting for different progeny classes, will alter the unbiased statistical changes in the distribution of heteroplasmy levels. Demonstration that a constraint existed would provide some foundation for the proposal that the "experimental evolutionary scheme" would have consequence-relief of this constraint. The crosses presented in this manuscript could have simply provided opportunity for random drift to fix an abundant pre-existing mutation. To make the claims regarding the "experimental evolutionary scheme", the manuscript would have to dismiss this null hypothesis and demonstrate the efficacy of the approach. At present, the appearance of one line out 12 with a mutant in the experimental sample and none in the 6 control lines is far from enough to argue that the outcrossing was essential to the recovery of the mutation.

The reviewer is correct in their argument that knowing the status of the heteroplasmy in the stock would have been important at the onset of the experiment. The best we can do is infer the heteroplasmy status in the stock at a time when we were evaluating and discovering the focal mtDNA mutation. We would be quite happy to characterize additional flies from the ancestral stock except that recognizing that this characterization is occurring >50 generations after the beginning of our experiment, the main point we wished to emphasize is that there was heteroplasmy in the original stock. We hope elimination of sentences regarding the transient nature of the heteroplasmy will minimize any confusion and avoid us overstating that we understand the dynamics of this particular mutation. While we agree that following the dynamics of this heteroplasmy would be very interesting, and would help bolster our claim of a “constraint-release” afforded by the experimental evolution scheme, such detailed characterization is beyond the scope of our paper. Even such a conclusion would be limited by the fact that the mutation became fixed in only 1 out of 12 experimental lines. Instead, we focus the bulk of the manuscript on what makes the mutation specifically male-harming.

*The “experimental evolutionary scheme" was designed to bypass the influence of indirect counter selection due to reduced fertility of the males: but to what extent does such a selection influence outcome? It appears that the heteroplasmic male flies are fertile (not tested, but failure to see fertility depression in lines with high levels of mutant genomes suggest that mutant genomes can be carried without a phenotype) and hence that no such counter selection would operate early after the appearance of the mutant allele (when it is rare and heteroplasmic). The persistence of mutant flies (perhaps heteroplasmic, but untested) in three of the six coevolving/control lines suggests the absence of a significant counter selection preventing their maintenance. Consequently, male sterility could only come into play, if at all, very late after the mutation rose to be a dominant allele with frequent production of homoplasmic derivatives, and since the male sterility phenotype is weak at the temperatures used for propagation, it would not be surprising that secondary selection due to male fertility could prevent fixation of the mutant. Furthermore, since the scheme did not use unrelated males for outcrosses but used males from the starting population, which apparently harbored the mutant for many generations (many generations are essential to produce homoplasmic flies), these males might already have carried suppressors of the phenotype. Thus, I do not think that manuscript offers support for claim that an effective scheme for recovery of male-harming mutations is described. I would like to see direct evidence that it is effective and efficient.*

The reviewer makes several good points here, which we address on-by-one. First, we can conclude with significant confidence that there are no suppressors of the sterility phenotype in the ancestral stock used to backcross. This is the same stock we did all our experiments to characterize the infertility and any suppressors would have been immediately evident. This is in spite of the fact that the mutation was present in this stock.

Second, the reviewer is doubtless aware that the presence of a cytoplasmically inherited male-harming mtDNA mutation would still be tolerated in the presence of the wildtype allele since only indirect selection acts on the mutant mtDNA. Nevertheless, we completely agree with the reviewer that the scheme we have proposed cannot be deemed essential for obtaining the COII^G177S^ mutation. We also agree with the reviewer that our findings show that the phenotype of COII^G177S^ only manifests when present at high levels of heteroplasmy, and that such penetrance issues will affect the detailed design of any experimental evolution strategy. To address the reviewer concerns, we have greatly de-emphasized the significance of the experimental evolution strategy and instead focus on the significance of finding and characterizing a male-harming mtDNA mutation.

The second component of the manuscript, the description of the phenotype of the male sterile mutation is detailed but I was not impressed that it gave us much insight into the defects and their connection to the mutation. One obfuscating issue is the use of high temperature to enhance the phenotype of an otherwise weak expressivity of the mitochondrial mutation. The highest temperature at which flies can be cultivated is generally limited by male sterility (e.g. David JR et al., 2005). For laboratory stocks, this highest temperature is usually taken as 29°C but populations tend to show differences in the temperature range across which they will grow-perhaps not surprisingly as environmental variables and adaptation would select for different ranges. At the temperature limit, defects in sperm development are usually seen as responsible for sterility (e.g. Rohmer C, et al., 2004). Interactions between nucleus, mitochondria and temperature have been described as affecting male sterility (Hoekstra et al., 2013). The manuscript shows an interaction of temperature and the mitochondrial mutation, but the study of the phenotype leaves an uncertainty whether the phenotype described is primarily that of the mutant or that produced at high temperature. That is, one might imagine that the mutation might sensitize the flies so that the high temperature defect shows up at 29°C, or one can imagine that 29°C increases the expressivity of the mutant phenotype, or something in between. In this regard, it would be nice to know if the sterile phenotype of w1118 wild type flies when raised at temperatures beyond 29°C, resembles or is distinct from the phenotype as seen in the mutant at 29°C.

We take advantage of the fact that the COII^G177S^ mutation affects male fertility in both age- and temperature- dependent fashion. We now show that the age-dependent infertility (performed at 25°C) is exactly what we would predict using additional cytological analysis of seminal vesicles, which we have shown is an accurate surrogate for male fertility. We find that the seminal vesicle size of COII^G177S^ mutant males is reduced compared to those with wildtype mtDNA, similar to what we found with young males raised at 29°C. Moreover, while *w1118* males with wildtype mtDNA also show impairment at 29°C, their seminal vesicle sizes are not significantly reduced. Our additional experiments show that the infertility defect cannot be solely explained as a confounding factor with temperature, and that the temperature-dependent reduction of fertility in wildtype *w1118* males is distinct from COII^G177S^-dependent defect at 29°C. We hope that our additional experiments should satisfy the reviewer’s concerns regarding this criticism. Moreover, we must emphasize again that this is, to the best of our knowledge, the most detailed characterization of the phenotypic consequences of multiple life-history traits (aging, fertility, neural function) in young and old males and females, at two different temperatures.

*The analysis of nuclear interactions is potentially exciting, but the reported analysis represents a first step. The loci responsible are not mapped or otherwise identified. Sequenced strains are analyzed but no mention is made of any finding emanating from this. I worry that modification might be the result of modification of temperature sensitivity. The things underlying this concern are that the maximal temperature of male fertility commonly vary between strains, genetic backgrounds that modified the reported phenotype are remarkably common, and the data shown for 29°C in the supplement of Figure 9 shows that w1118 is unusually weak even at the temperature selected as widely permissive. If the temperature is raised to say 31°C, the wild type w1118 stock is likely to be male sterile: I would like to know whether temperature tolerance is increased (i.e. this male sterility expected for the wild-type w1118 strain is suppressed) by out crossing of the wild type to the same strains that suppressed the mutant. This control might reveal that the recorded affects are due to expected widespread differences in thermo-tolerance. This concern that the modification may be due to a more generic affect is partly alleviated by the reported increase in Cox activity in suppressed mutant backgrounds.*

The main conclusion we wish to draw from the nuclear suppressor findings is that there is a significant nuclear genetic variation to suppress the male-harming effects of mtDNA mutations like COII^G177S^. We seriously considered the reviewer’s criticism that this could also be a pleiotropic effect due to the high temperatures we used to fully manifest the male infertility. Once again, using aged males, we show that the heterozygous OregonR nuclear background completely rescues (old) male fertility defects at 25°C that are caused by the COII^G177S^ mutation and that this fertility rescue is associated with suppression of the decrease in seminal vesicle size. Together with our earlier presented data, our findings strongly suggest that the OregonR nuclear genome harbors a ‘true’ suppressor of the COII^G177s^-dependent male fertility defect and its mechanism of suppression is likely the same at 25°C and 29°C. We hope that our revised manuscript emphasizing our detailed characterization of how a hypomorphic mtDNA mutation can cause male-specific impairment will be viewed favorably by the reviewer. While we share the reviewer’s excitement about the potential mapping of these nuclear suppressors, we point out that this is not a trivial undertaking. The first author hopes to identify and characterize the nuclear suppressor mutation(s) in a future study which will take at least a few more years.

[Editors' note: the author responses to the re-review follow.]

Summary:

In their final submission the authors should address better previous criticism influenced by the fact the supposed approach to isolate male harming mutants was the most innovative part of the original submission. The new version is hence weaker, but still a nice report and should be published at eLife. Of two previous criticisms, the bigger one is that the data are not sufficient to document homoplasmy. While this could be serious flaw, it is very likely that the line is close to homogeneous and if it were not and the authors sampled flies that were heteroplastic in their tests of fertility they would have seen phenotypic variation. If the authors answer in this way, and put more guarded statements in their discussion, we would be satisfied on this account. The are other issues, on the phenotypic analysis which are more about making things more clear, particularly aspects that are buried in the supplementary data. Eliminating some unfounded discussion will bring more sharpness, but this is something that the authors are best placed to address. These are detailed below.

With new experiments (detailed below), we now show that the strains we analyzed are near homoplasmy (100% or 99.9%). We also address the other concerns raised by the reviewers.

*Essential revisions:*

*The presentation of the data has been changed substantially, while the data remain much the same. The change in the presentation has resolved one of the central concerns about the previous version of the manuscript that presented a crossing scheme as an effective selection for male harming mutations. The critiques, certainly mine, argued that this was not supported by the data and that the mitochondrial mutation preexisted and was uncovered by chance. The present draft still argues that this scheme should be effective, which I severely doubt based on pragmatics of mutation frequency and infrequent segregation of homoplasmic lines. I think that the unsubstantiated sentence in the current version saying that "we reasoned that multiple mutations per mtDNA would be sampled in 35 generations across 18 lines" should be eliminated or justified.This is completely untenable as shown by the mathematical analysis of the number of generations required for lines to progress to homoplasmy (See Solignac). Nonetheless, the major change in presentation has corrected the biggest problem with the previous submission.*

The reviewers express concern over our statement that multiple mtDNA mutations might be sampled over 35 generations across 18 lines. Note that we intentionally used the word ‘sampled’ to distinguish mutations that arise versus those that arise and drive to homoplasmy. In addition, while mtDNA segregation can take many generations to reach homoplasmy, dramatic instances of rapid shifts in heteroplasmy have also been observed in humans, cows, and flies (Brown et al., 2001; Hauswirth & Laipis, 1982; Olivo et al., 1983; Matsuura et al., 1990; Matsuura et al., 1991; Koehler et al., 1991; Jenuth et al., 1996; Matsuura et al., 1997; Sato et al., 2007; Fan et al., 2008; Lee et al., 2012; Freyer et al., 2012; Sharpley et al., 2012; Ma, Xu & O’Farrell, 2014; Ma & O’Farrell, 2015; Ma & O’Farrell, 2016). Nevertheless, to avoid any ambiguity, we now state that this period may not be necessary to drive mtDNA mutations to near homoplasmy and now cite the Solignac reference (Solignac et al. Genetics 1987).

The question now is, what is the merit of the present manuscript. The authors do report a maternally inherited partial male sterility trait that is linked with an SNP mt:CoII. The recognized SNP may well be the cause of the phenotype, but the possibility of linked changes in the unsequenced 4kb regulatory region was not eliminated (note that this is the region that changes the most rapidly and spontaneous variants of this region have been shown to arise in laboratory stocks – see Rand). Furthermore, identification of a mitochondrial DNA mutation altering an electron transport component that causes male sterility is not entirely novel (see Xu). The manuscript does report a fairly detailed effort to characterize the defect and shows suppression of this defect in crosses that introduce different nuclear backgrounds. In this regard, it goes beyond other reports that argued for nuclear modification of phenotypes caused by mitochondrial mutations. In our opinion these findings are of interest and could constitute a significant advance. We are however, left with some hesitancy because the merit ought to lie in the rigor in of the documentation of the principles, and we see some shortcomings.

The reviewer raises two points. First, there is a possibility of linked changes in the unsequenced 4 kb regulatory region- the so-called D loop. We now explicitly suggest this possibility in the Discussion. However, we would like to point out that the causal connection with the COII^G177S^ mtDNA mutation is quite strong. We have observed a biochemical defect in cytochrome C oxidase activity that correlates perfectly with the male infertility phenotype. This COX activity is rescued in the suppressor background – these data strongly suggest that the G177S mutation causes the COX defect, which in turn causes the fertility defect. This is as strong a statement that can be made with a causal connection with a single mtDNA mutation, the exception being the Xu et al. paper that used a restriction enzyme strategy to introduce mtDNA mutations. In all other cases that have mapped a phenotype to a single mutation in mtDNA they did not assay or sequence the A+T region (Camus et al. 2015, Meiklejohn et al. 2013, Clancy et al., 2011, Xu et al. 2008, and Clancy 2008).

Second, the reviewer is correct that the *mt:CoI^R301L^*mutant in a previous report (Xu, DeLuca & O’Farrell Science 2008) was also associated with male sterility. However, this report did not quantitatively assess other aspects of male or female fitness. We now more fully discuss the *mt:CoI^R301L^*mutant in our revisions, pointing out the biochemical and biological differences between this mutant and the one we characterize. Even though both mutants had a male sterility defect (*mt:CoI^R301L^*was completely male sterile), the authors observed no defects in cytochrome oxidase activity and spermatogenesis. Instead, the infertility was reported due to a defect in sperm storage. Thus, we have a unique manifestation of a male-harming mtDNA mutant. Additionally, we believe that our claim regarding the specificity of the G177S mutation in causing male sterility is based on one of the most comprehensive and quantitatively rigorous phenotypic characterizations.

*One concern is that the data are not sufficient to rigorously demonstrate the genetic homogeneity of the lines examined. The details are below, but this view can be summarized as follows: the original plan treated the crosses as if they were a selection that would yield a pure male harming mutation, and suitable measures where not take to ensure the genetic homogeneity of the stock that was retrieved. Given the number of mt genomes carried by flies, a high sensitivity of detection of a rare allele (in a range approaching 0.1%) and analysis of single flies is needed to rigorously demonstrate homoplasmy.*

We believe that the reviewers’ concerns can be remedied by clarified wording and one critical additional experiment that we have performed with the help of new co-author Scott R. Kennedy (now added). First, we now clarify in the text that subsequent to identification of the G177S mutation in experimental line 7, all experiments were done from a mutant and wildtype mtDNA stocks reestablished from the original w118 strain. Additionally, we have now sequenced pools and individuals of these stocks to show that the pools of flies are nearly homoplasmic:

“We characterized the nature of the heteroplasmy in the re- isolated COII^G177S^ mutant mtDNA lines by sequencing a pool of COII^G177S^ mutant mtDNA and a separate pool of re-isolated wildtype mtDNA adult fly heads using a Duplex Sequencing strategy (Kennedy, Schmitt et al. 2014) (Figure 4—figure supplement 1). […] We therefore used these re-isolated strains for all subsequent phenotypic analyses.”

*A second issue concerns data on fertility demonstrating that in an early first mating the males exhibited no (at 25°C) or weak (29°C) deficits in fertility. At least in reading the text, the subsequent analyses did not seem to take this fully into account. Many of my concerns here could be dealt with by better reference to supplementary material and more careful wording. Nonetheless, it still seemed to me that this issue was not fully investigated and the impact of this finding on other analyses was not always considered.*

All subsequent analyses to document infertility or its suppression were done in either aged males at 25°C or young males at 29°C; in both instances the deficit in fertility is robustly observed. Nevertheless, we have made changes to make sure that all references and caveats are clearly cited in our Results and Discussion.

*Overall, this an interesting report that suffers from weakness in its details. These can be addressed in the revised final submission*

We are grateful for the opportunity to suitably address these concerns and those listed below. We believe that the clarifications and the additional experiments regarding the status of heteroplasmy should satisfy many of the reviewer concerns.